# Extrinsic mechanical forces mediate retrograde axon extension in a developing neuronal circuit

M.A. Breau[1,2], I. Bonnet [2,3], J. Stoufflet[1,2], J. Xie[1,2], S. De Castro[1,2] & S. Schneider-Maunoury[1,2]

To form functional neural circuits, neurons migrate to their final destination and extend axons towards their targets. Whether and how these two processes are coordinated in vivo remains elusive. We use the zebrafish olfactory placode as a system to address the underlying mechanisms. Quantitative live imaging uncovers a choreography of directed cell movements that shapes the placode neuronal cluster: convergence of cells towards the centre of the placodal domain and lateral cell movements away from the brain. Axon formation is concomitant with lateral movements and occurs through an unexpected, retrograde mode of extension, where cell bodies move away from axon tips attached to the brain surface. Convergence movements are active, whereas cell body lateral displacements are of mainly passive nature, likely triggered by compression forces from converging neighbouring cells. These findings unravel a previously unknown mechanism of neuronal circuit formation, whereby extrinsic mechanical forces drive the retrograde extension of axons.

[1] Institut de Biologie Paris-Seine (IBPS)—Developmental Biology Laboratory, CNRS UMR7622, INSERM U1156, F-75005 Paris, France. [2] Sorbonne Universités, UPMC Univ Paris 06, 75005 Paris, France. [3] Laboratoire Physico Chimie Curie, Institut Curie, PSL Research University, CNRS UMR168, 75005 Paris, France. Correspondence and requests for materials should be addressed to M.A.B. (email: marie.breau@upmc.fr)

Neuronal networks are the functional building blocks of the nervous system. Their formation requires the movement of neurons towards their final location, where they establish functional connections with target cells. In the peripheral nervous system, sensory neurons gather from an initial spread distribution of cells to form compact structures: dorsal root ganglia assemble from migrating streams of mesenchymal neural crest cells (NCCs) in the trunk[1], while the progenitors of cranial ganglia and sensory organs coalesce from large regions of the pan-placodal domain (reviewed in refs. [2–6]). Neuronal clustering has numerous potential roles in sensory development and function: it may be essential for axons to use common navigation cues or interact with each other to establish neural maps[7], and for somata to integrate sensory inputs when the circuits are functional. Sensory neurons have not only to find their position in the neuronal cluster, but also to form axons that extend towards and penetrate into the brain or spinal cord at discrete entry points. Contacting these intermediate targets is crucial for appropriate innervation of final target regions in the central nervous system. Despite some insights into the molecular pathways involved, little is known about the cellular dynamics underlying the clustering of cranial sensory neurons and the formation of sensory axons and their contact with entry points on the brain surface[8–15]. Even less is known about whether and how these two processes are coordinated in vivo.

Here, we use the zebrafish olfactory placode (OP) as a model system to address the underlying mechanisms. At 24 hpf (hours post fertilisation), the two OPs are spherical clusters of neurons that project fasciculated axons towards the olfactory bulb in the anterior brain (telencephalon). OPs assemble from two elongated cell fields surrounding the brain, which coalesce into paired compact spherical clusters between 15 and 21 hpf[16], through yet undescribed morphogenetic movements. In the olfactory circuit, neurons are born in two waves. A transient population of pioneer neurons differentiates first, during morphogenesis of the cluster. Their axons have been seen elongating dorsally out of the placode, along the brain wall, at 20 hpf[17, 18]. Pioneer axons are then used as a scaffold by later born olfactory sensory neurons to outgrow their axons towards the olfactory bulb[17]. Although chemical cues guiding the navigation of zebrafish olfactory axons in the brain domain have been identified[8, 19, 20], how axons form and elongate within the OP territory remains unknown.

We use multiscale quantitative imaging to dissect out the mechanisms underlying OP morphogenesis and the formation of the first axons to contact the brain. Our data show that active convergence movements along the brain coordinate with passive lateral displacements of cell bodies away from the brain to sculpt the final OP cluster. Surprisingly, axonal protrusions form during lateral movements, through a non-canonical mechanism referred to as retrograde axon extension, whereby somata move away from axon tips attached to the brain wall at the location of the entry point. Cell nucleus deformation patterns and laser ablation experiments further suggest that actively converging cells coming from placode extremities exert compressive forces in the placode centre that squeeze out central neurons from the brain surface, thus contributing to the elongation of their axons. Our findings unravel an unexpected mechanism of neuronal circuit development, where extrinsic mechanical forces drive retrograde axon extension, a wiring strategy that could account for neuronal circuit formation in other regions of the nervous system.

## Results

### OP morphogenesis does not require apoptosis or cell division.
Fate map experiments showed that morphogenesis of the paired OPs occurs by the transformation of two stripes of cells into spherical clusters between 15 hpf (or 12 somites, 12 s) and 21 hpf (or 24 s; Fig. 1a, b)[16]. Despite recent imaging efforts[21, 22], we still lack a high-resolution analysis of cell behaviours during this morphogenetic event. We took advantage of the *ngn1:gfp* line[23] that labels a subpopulation of OP cells known as the early-born neurons, which include pioneer neurons[18]. At 12 s, the *ngn1:gfp* line labelled two elongated cell groups on either side of the brain, two to three cells wide and one to two cells thick (Fig. 1c, d, *left panels*), and expressing Dlx3b (Supplementary Fig. 1a), a marker for OP cells[21, 22]. At 24 s, *ngn1:gfp*+ cells formed paired ellipsoidal clusters next to the brain. As previously observed[18], an almost total overlap between green fluorescent protein (GFP) and the neuronal marker HuC could be seen at 24 s (Supplementary Fig. 1b), confirming that *ngn1:gfp*+ cells represent the OP early-born neurons.

Several cell behaviours can be implicated in changes of tissue shape, including apoptosis, proliferation, cell size or morphology changes, and cell movements or rearrangements[24–27]. During OP morphogenesis, very few apoptotic cells were detected, and inhibition of apoptosis had no effect on final placode shape (Supplementary Fig. 2). OP GFP+ cells divided during morphogenesis, with a uniform spatial distribution of cell divisions in the placode (Supplementary Fig. 3a), and pharmacological perturbation of cell proliferation did not affect OP morphogenesis (Supplementary Fig. 3b–d and Supplementary Movie 1). Using three-dimensional (3D) cell reconstruction, we observed that cells had a smaller volume at the end of morphogenesis (24 s) than when clustering starts (12 s), resulting from a smaller cell body in the XY plane, rather than a shortening in the Z orientation (Supplementary Fig. 3e–g). This could be due to contraction of cell bodies in the XY plane[26] and thus contribute to OP coalescence. However, following the nucleus size over time as readout for cell size (see the correlation between the two parameters in Supplementary Fig. 3h), we showed that the decrease in cell size results from cell division rather than cell contraction (Supplementary Fig. 3i, j). Consistently, treatment with drugs blocking proliferation led to OPs of normal volume containing bigger cells (Supplementary Fig. 3k, l). Collectively, these experiments rule out a major implication of cell death or of cell proliferation and associated cell size reduction in OP morphogenesis. We therefore focused our attention on cell movements.

### Convergence and lateral cell movements shape the OP.
To analyse cell movements, we performed live imaging on *ngn1:gfp* transgenic embryos (N = 10 embryos). As placode assembly proceeded, the *ngn1:gfp*+ elongated cell fields progressively shrunk along the anteroposterior (AP) axis, while getting larger along the mediolateral (ML) axis, thus acquiring their final rounded shape (Fig. 1a–d, Supplementary Fig. 4a and Supplementary Movie 2). The cell group also thickened slightly over time along the DV axis (Fig. 1d). Taking advantage of nucleus or membrane red labelling, we tracked individual *ngn1: gfp*+ placodal cells in 3D, throughout the whole morphogenesis process (n = 83 cells, Supplementary Movies 2 and 3). This analysis revealed that placodal cells moved towards the future position of the placode, in the central region of the initial field, and lateral to the brain (Fig. 1e). Most cells from the anterior and posterior regions of the initial placodal field displayed a 'two-phase' trajectory: they first moved parallel to the brain surface to converge towards the centre of the placode (AP or convergence movements), then moved laterally, away from the brain (ML or lateral movements; Fig. 1f). Central cells underwent lateral movements only, without drastic displacement along the AP axis (Fig. 1e, f). All these movements were persistent and directional, as shown by the mean square displacement (MSD) analysis of the tracks (Fig. 1g). To assess whether cells exchange neighbours during their movements, we tracked pairs of

*ngn1:gfp+* neighbours and followed small groups of Kaede photoactivated cells during OP morphogenesis (Supplementary Fig. 5). This revealed that the displacements of initial neighbours were coordinated through time, with a short-range intermixing (Supplementary Fig. 5), likely resulting from cell division or local neighbour exchange during movements. Altogether, we conclude

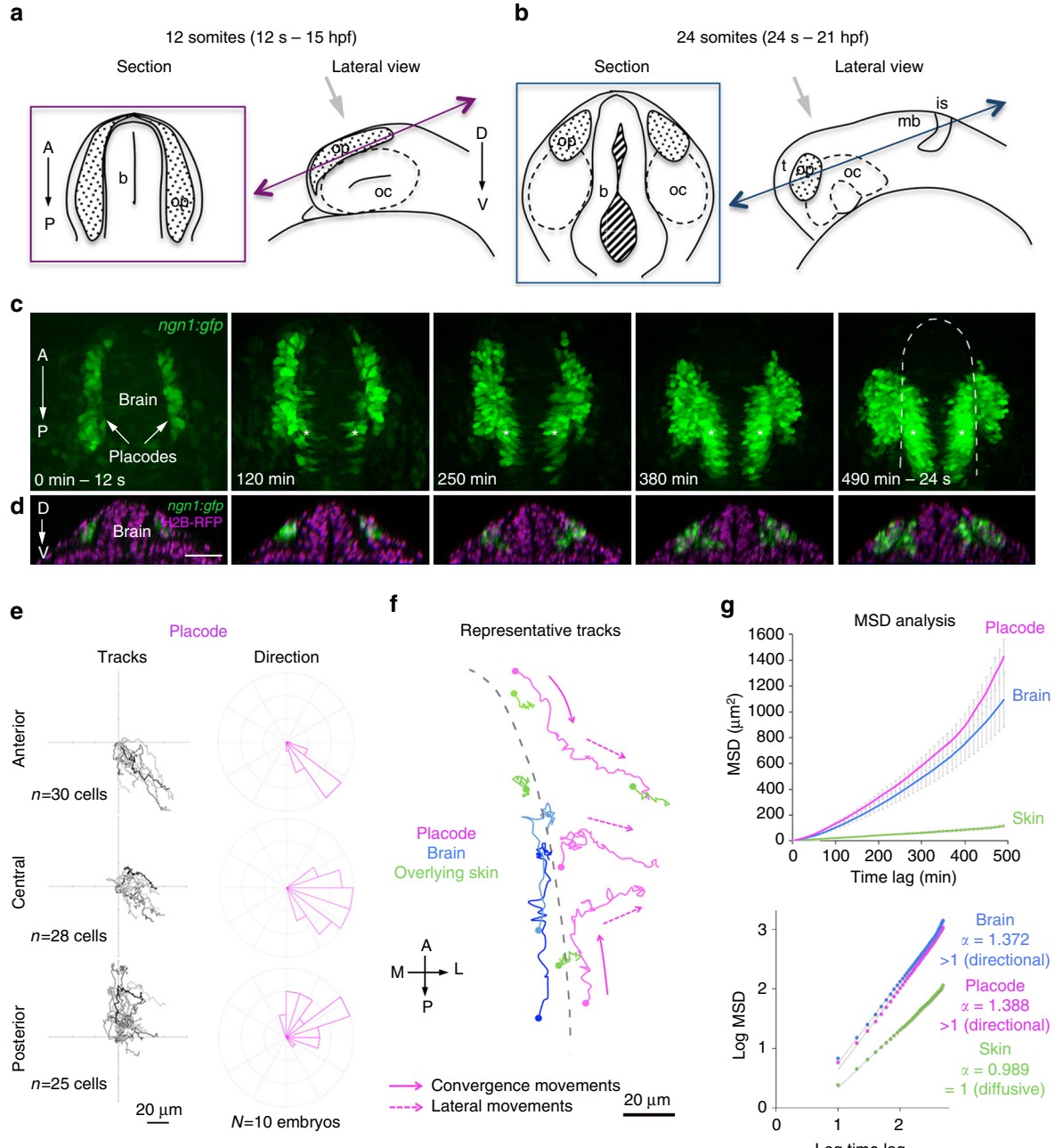

**Fig. 1** Quantitative live imaging analysis of cell movements during OP morphogenesis. Live imaging of OP morphogenesis was performed between the 12 somites (12 s) and 24 s stages. **a**, **b** *Schematic views* of the head regions of 12 s **a** and 24 s **b** embryos. For each stage, the *right panel* shows a lateral view (orientation of the microscope objective indicated with a *grey arrow*). The *left panel* represents the optical section indicated by *double arrows* in lateral views. *b* brain, *is* isthmus, *mb* midbrain, *oc* optic cup, *op* olfactory placode, *t* telencephalon. **c** Live imaging on a *ngn1:gfp* embryo between 12 and 24 s, showing the progressive coalescence of the two elongated GFP+ OP domains into compact and spherical clusters on each side of the brain (*XY dorsal view of the head*, maximum projection of a 92 μm Z-stack). *Asterisks* indicate GFP expression in the brain. **d** *YZ* sections corresponding to the images shown in **c**, with H2B-RFP-labelled nuclei (*magenta*), showing the shape of the brain and placode tissues along the DV axis. *Scale bars*: 50 μm. **e** Tracks of anterior, central and posterior placodal cells (as defined in Supplementary Fig. 4b), merged at their origin, and associated directions of movement (as defined in Supplementary Fig. 4c). All cells were tracked throughout the morphogenesis process, during a 500 min period of time. **f** Representative tracks of placodal cells from three different AP positions (*magenta*), overlying skin cells (*green*) and adjacent brain cells (*blue*). Skin cells are located above the brain and the placode. *Dots* represent initial positions. The *dotted grey line* indicates the brain surface at 12 s. Cells from placode extremities move parallel to the brain surface to converge towards the centre of the placode (convergence movements along the AP axis, *full arrows*), then move laterally, away from the brain (lateral movements along the ML axis, *dotted arrows*). Central cells undergo lateral movements only. **g** MSD plot and its log equivalent for placode, skin and brain cell trajectories. α is the slope of the *log plot* and is used an indicator of directional (>1) vs. diffusive (=1) movement

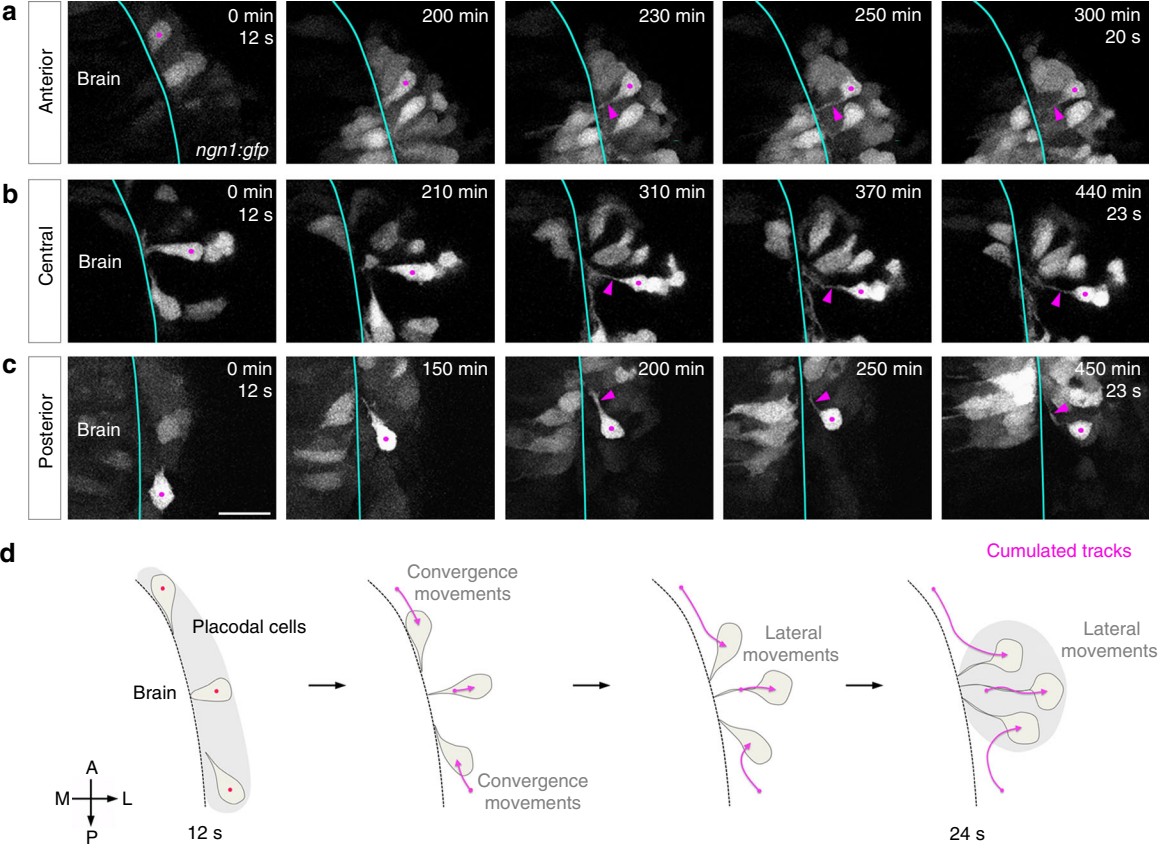

**Fig. 2** Analysis of cell morphologies during OP morphogenesis. Images extracted from movies on stable **a**, **c** or transient **b** *ngn1:gfp* transgenic embryos, showing the movement and morphologies of anterior **a**, central **b** and posterior **c** OP cells. *Magenta dots* label cell bodies of neurons of interest (strongly GFP+) and *magenta arrowheads* point to their protrusions contacting the brain. *Scale bar*: 25 μm. **d** Schematic representation of the cell morphologies observed during OP morphogenesis for cells from different AP positions, and the associated cumulated tracks (*magenta arrows*). *Grey regions* represent the initial and final shapes of the OP. All images represent dorsal views

that a combination of convergence and lateral cell movements drives OP morphogenesis.

Placodal cells could passively follow large-scale movements of adjacent tissues. Indeed, a previous study showed that, during early somitogenesis, OP and telencephalic cells undergo concerted anteriorward movements[22]. We compared the motion of placodal cells with that of other cell types in their vicinity, namely *ngn1:gfp*+ cells from the adjacent brain (*n* = 38 cells) and cells of the overlying skin (*n* = 39 cells), from 12 s onwards (*N* = 10 embryos). Skin cells moved on short distances without preferential direction (Fig. 1f and Supplementary Fig. 4d, f) and exhibited a diffusive-like motion (Fig. 1g), whereas GFP + brain cells underwent anteriorward directional movements (Fig. 1f, g and Supplementary Fig. 4e, f), with trajectories that were not correlated to those of placode cells except partially in the posterior region (Supplementary Fig. 4g). Thus, the movements of OP cells do not simply follow those of cells from surrounding skin and brain tissues, but rather have their own specific dynamics.

**Axons form during lateral movements by retrograde extension.** To determine when and how the first axons form, we used mosaic labelling to analyse individual cell morphologies during the movements shaping the placode. To do so, we took advantage of differences in GFP levels in *ngn1:gfp* stable embryos, or mosaic *ngn1:gfp* expression obtained by DNA injection or cell transplantation. During convergence movements, anterior and posterior cells exhibited drop-like morphologies, with short

protrusions oriented towards the direction of their movement (Fig. 2a, c, d). The protrusions most often made dynamic contacts with the brain surface, which got stabilised when cells arrived in the placode centre. Then started a second phase of movement: while the protrusions kept contact with the brain, cell bodies moved laterally in the placode (lateral movements), away from the brain, thereby elongating thin processes in their trailing edge (Fig. 2a, c, d, see also Fig. 3c). Cells initially located in the placode centre formed similar trailing protrusions during lateral movements (Fig. 2b, d, see also Fig. 3h). Mosaic expression of a membrane-targeted mCherry (mbCherry) in *ngn1:gfp* stable embryos showed that this feature is not restricted to cells expressing high GFP levels, as low GFP cells exhibited a similar cellular morphogenesis (Supplementary Fig. 6).

During lateral movements, the protrusions contacted each other to form a bundle in the centromedial region of the placode, juxtaposed to the brain wall. The bundle of GFP + protrusions could be seen from 18 s onwards in all our movies performed on *ngn1:gfp* stable embryos (Fig. 3a, b and Supplementary Fig. 6a). On the basis of their shape and attachment to the brain, we hypothesised that these protrusions are the axons of early-born OP neurons. To test their axonal identity, we first analysed whether their cytoskeleton composition exhibits axonal characteristics. The shaft of the protrusions contained microtubules (Fig. 3c, d). Live imaging of microtubules with Doublecortin–GFP showed that elongation of the protrusions during lateral movements coincided with an extension of the microtubule backbone (Fig. 3c and Supplementary Movie 4). In contrast, actin, visualised with the

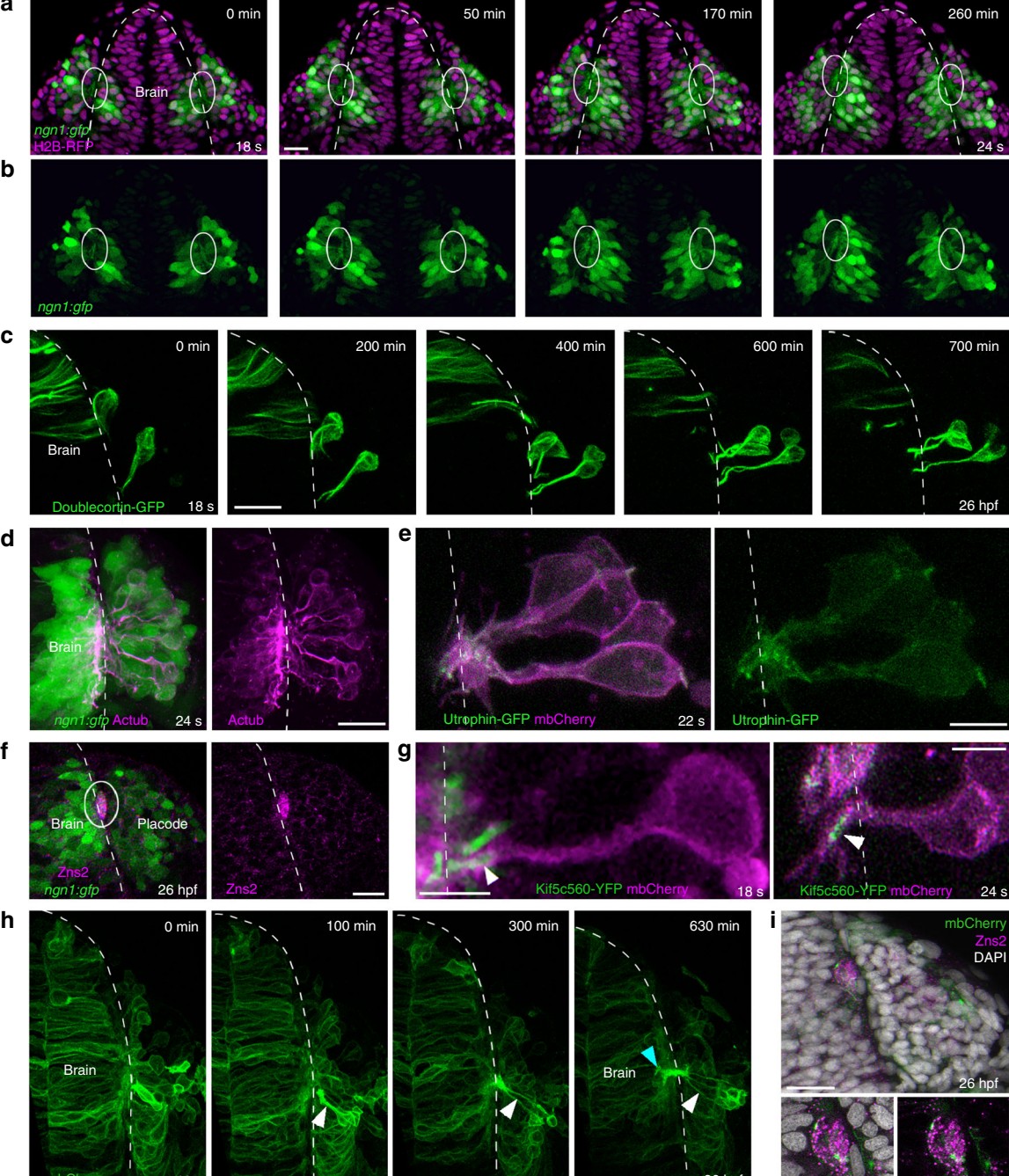

**Fig. 3** Axonal identity of the cytoplasmic protrusions. All images represent dorsal views; *dotted white lines* indicate the brain surface. **a**, **b** *XY* sections extracted from a movie performed on a *ngn1:gfp* embryo injected with H2B-RFP mRNA. From 18 s onwards, the cytoplasmic processes connecting cell bodies to the brain form a bundle of GFP+ protrusions juxtaposed to the brain surface, in the ventromedial region of the placode (framed with *white lines*). **c** Live imaging on a wild-type embryo transplantated with Doublecortin–GFP-expressing cells, showing microtubules in the shaft of the protrusions and around cell bodies during cell movements. **d** Acetylated tubulin immunostaining (*magenta*) performed on a *ngn1:gfp* embryo, indicating the presence of stable microtubules in GFP+ protrusions and cell bodies at 24 s. **e** Mosaic labelling of actin and membranes, obtained with a transplantation of Utrophin-GFP (*green*) and mbCherry (*magenta*) expressing cells in a wild-type embryo. **f** Immunostaining for the OP pioneer axon marker Zns2 (*magenta*) on a *ngn1:gfp* embryo, labelling the bundle of GFP+ protrusions at 26 hpf (*white lines*). **g** High magnification of OP cells expressing mbCherry and the axonal specification marker Kif5c560-YFP, showing the accumulation of Kif5c560-YFP at the tip of the protrusions (*arrowhead*) during lateral movements. On the *left*, the embryo was co-injected with Kif5c560-YFP mRNA and pCS2-mbCherry DNA, explaining why some of the Kif5c560-YFP + accumulations are not associated with *magenta* cells. On the *right*, a wild-type embryo was transplanted with cells co-expressing Kif5c560-YFP and mbCherry mRNAs. **h** Long-term live imaging of an embryo injected with mbCherry (*green*) mRNA, from 18 s to 26 hpf stages. A few OP cells express higher levels of mbCherry, which allows to visualise the elongation of their protrusions during lateral movements (*white arrowheads*), and their entry into the brain territory (*green arrowhead*). **i** Zns2 immunostaining (*magenta*) performed on the embryo imaged in **h**, showing the mbCherry+ protrusions within the Zns2+ bundle at 26 hpf. *Scale bars*: 25 µm in **a**, **c**, **d**, **f**, **h**, **i**, and 10 µm in **e**, **g**

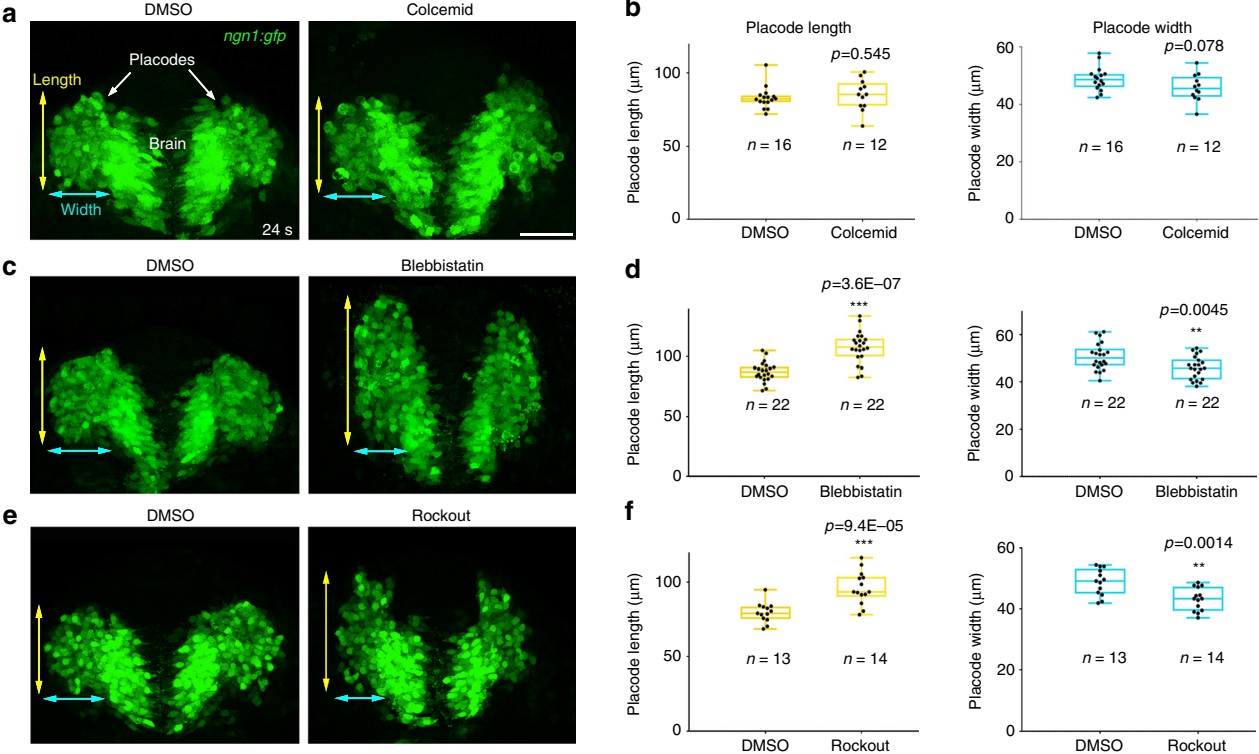

**Fig. 4** Effects of Colcemid, Blebbistatin and Rockout treatments on OP morphogenesis. **a** Colcemid-treated *ngn1:gfp* embryos do not show any defects in OP morphogenesis at 24 s, as compared with DMSO controls. **b** Quantification of OP length and width at 24 s, in colcemid-treated embryos and DMSO controls. **c** Blebbistatin-treated *ngn1:gfp* embryos exhibit longer and thinner OPs at 24 s as compared with DMSO controls. **d** Quantification of OP length and width at 24 s in blebbistatin-treated embryos and DMSO controls. **e** *ngn1:gfp* embryos treated with Rockout show morphogenesis defects that are similar to those of blebbistatin-treated embryos. **f** Quantification of OP length and width at 24 s in embryos incubated with Rockout or DMSO. *n* indicates the number of analysed placodes (one placode per embryo) in each condition. *p* values: unpaired two-tailed *t*-tests. *Scale bar*: 50 μm

Utrophin–GFP probe, was virtually absent from the protrusion shafts during lateral movement. It was rather located at the tip of the protrusions near the brain wall, where many filopodia-like structures explored the environment (Fig. 3e). Thus, the protrusions displayed axon-like cytoskeleton composition and growth cone morphology at their tips. To confirm their axonal identity, we analysed the expression of two axonal markers. We performed immunostaining for Zns2, a marker for OP pioneer axons[17], and saw its expression in the bundle of protrusions in close apposition to the brain (Fig. 3f). To support this result, we imaged live embryos expressing Kif5c560-YFP, an axonal specification marker[28, 29], and observed that it accumulates at the tip of the protrusions during lateral movements (Fig. 3g). In addition, long-term live imaging followed by Zns2 immunostaining showed the elongation of the initial protrusions dorsally out of the placode territory, along the brain surface and within the Zns2+ bundle (Fig. 3h, i and Supplementary Movie 5). Altogether, these results demonstrate that the long protrusions are the axons or future axons of the OP early-born neurons. Thus, in this context, axon extension occurs by displacement of cell bodies away from axon tips anchored to the brain surface, which contrasts from the textbook paradigm where axons outgrow from cell bodies and navigate towards their target[30, 31]. We refer to this non-canonical mechanism as retrograde axon extension.

**Convergence is active, whereas lateral movements are passive**. We next investigated whether the convergence and lateral cell movements that appear to organise the neuronal circuit are powered by intracellular activity of key cytoskeletal components involved in cell movements and axon extension in other contexts.

In OP cells, microtubules were present in the shaft of the axons and formed a cage around cell bodies (Fig. 3c, d and Supplementary Movie 4). To test their role in convergence and lateral cell movements, we treated *ngn1:gfp* embryos with colcemid, a drug that perturbs microtubule polymerisation[32]. Whereas cell division and acetylated tubulin staining were perturbed upon treatment (Supplementary Fig. 7a–c), colcemid-treated embryos showed no overt defect in OP morphogenesis (Fig. 4a, b), suggesting that convergence and lateral cell movements are not microtubule-dependent.

By contrast, perturbing the function of myosin II with drugs blocking its activity (blebbistatin) or that of its upstream activator Rock (rockout) significantly impaired OP morphogenesis: in drug-treated embryos, OPs were longer and thinner than in controls at 24 s (Fig. 4c–f), suggesting that both convergence and lateral cell movements are affected upon myosin II inhibition. Live imaging (Supplementary Movie 6) and individual cell tracking in drug-treated *ngn1:gfp* embryos (Supplementary Fig. 8) showed altered persistence and directionality in the movements of anterior and central placodal cells, but not posterior cells (Supplementary Fig. 8c–g). Brain cells behaved as in controls (Supplementary Fig. 8b, f, g) but, surprisingly, skin cells exhibited clear anteriorward movements in both drug conditions, in sharp contrast to the unoriented and diffusive behaviours of skin cells in controls (Supplementary Fig. 8a, f, g). While this analysis brings new information about the phenotype observed upon drug-mediated global inhibition of myosin II (Supplementary Fig. 8g), the affected skin cell movements preclude to conclude about the placode or cell autonomy of the defects.

In order to assess the cell autonomy of OP cell movements, we first used transplantation experiments to achieve mosaic

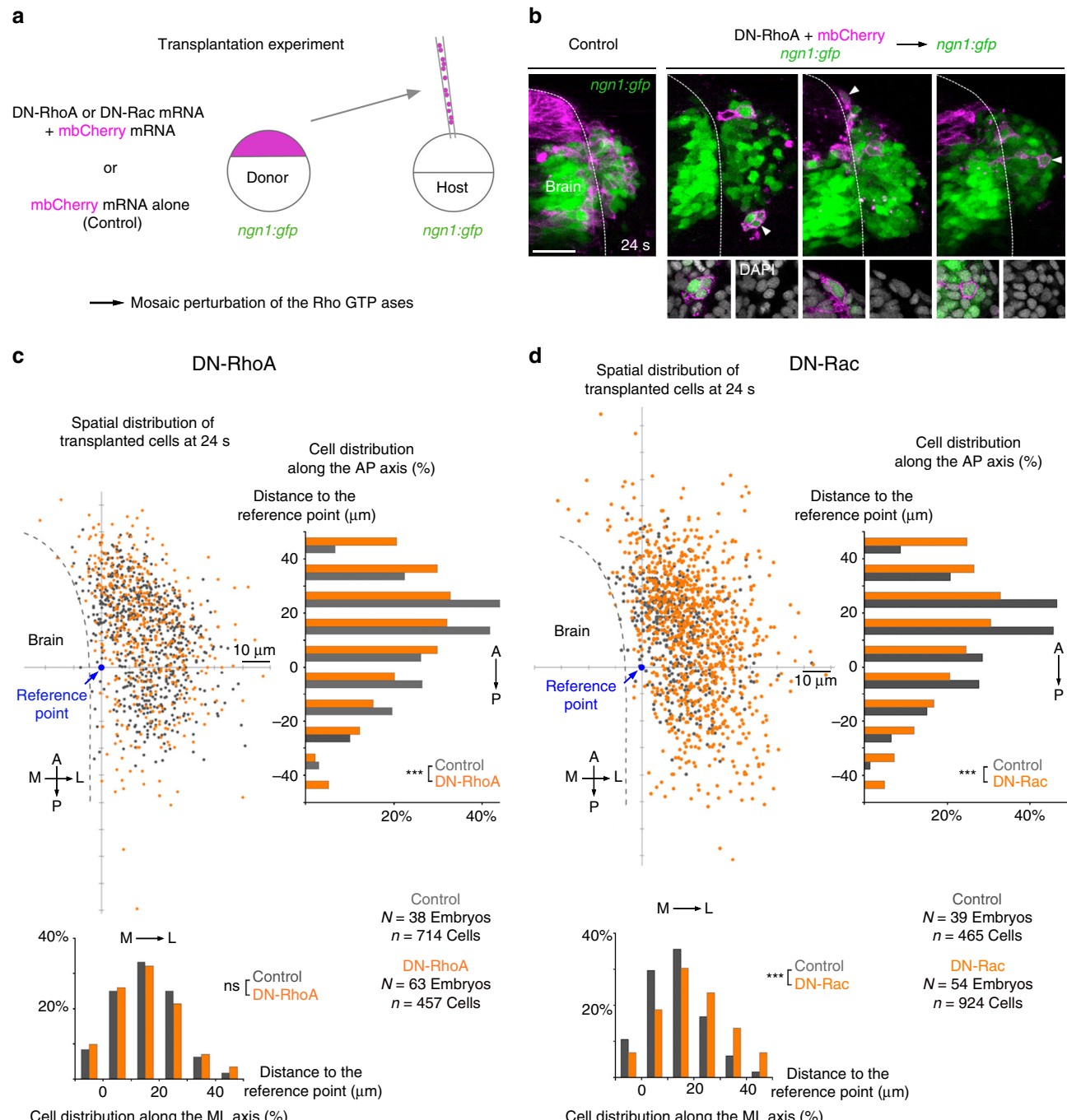

**Fig. 5** Mosaic perturbation of RhoA and Rac function. **a** Transplantation experiment set-up: cells from *ngn1:gfp* donor embryos co-expressing mbCherry and a dominant-negative form of RhoA (DN-RhoA) or of Rac (DN-Rac), or mbCherry alone (controls) were transplanted into *ngn1:gfp* host embryos, in order to achieve a mosaic perturbation of RhoA or Rac function. **b** Examples of embryos showing transplanted cells spanning the whole OP in controls, and instances of ectopic posterior and anterior transplanted cells expressing DN-RhoA. The *right panel* shows a DN-RhoA+ cell occupying a lateral position in the host placode. *Insets* show individual cells or cell groups expressing DN-RhoA, co-stained with DAPI to show that these cells are alive. *Scale bar*: 50 μm. **c** Spatial distribution of transplanted cells at 24 s, in control (*dark grey*) and DN-RhoA (*orange*) conditions, and projections of the spatial distribution along the AP and ML axis. The reference point is defined as the position of the axon bundle on the brain surface. $\chi^2$-tests are used to identify statistically different distributions (***$p < 0.001$). DN-RhoA+ cell distribution is more spread than that of control cells along the AP axis, but not along the ML axis, showing that DN-RhoA cell autonomously affects AP distribution of cells, but not their lateral dispersion. **d** Same analysis as in (**c**) with DN-Rac conditions in *orange*. Distributions are statistically different both along the AP axis and along the ML axis. As for DN-RhoA, DN-Rac+ cell distribution is more spread along the AP axis than in controls. Along the ML axis, DN-Rac+ cells distribute further laterally as compared to control cells, suggesting more efficient lateral movement

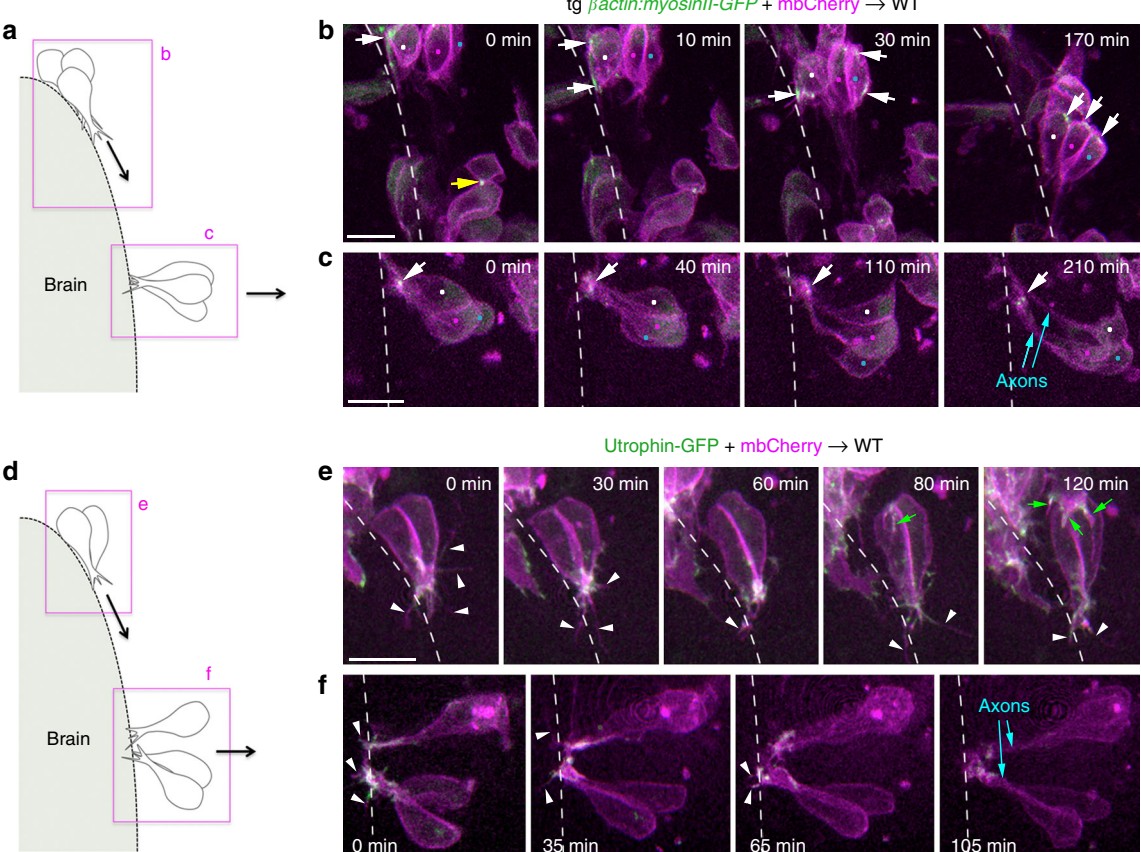

**Fig. 6** Actomyosin dynamics and protrusive activity during convergence and lateral cell movements. **a** Schematic view of cells imaged in **b**, **c**. Wild-type embryos were transplanted with cells from a *ßactin:myosinII-GFP* transgenic donor injected with mbCherry mRNA. **b** Anterior OP cells undergoing convergence movements towards the placode centre. Dynamic accumulations of myosin II can be observed in the cell bodies, in the front or back of moving cells (*white arrows*). The *yellow arrow* indicates myosin II accumulation during ring contraction after a cell division. **c** Central OP cells undergoing lateral movements. No myosin II can be detected in the cell bodies. Myosin II rather accumulates at the tip of the axonal protrusions (*arrows*). *Coloured dots* indicate the cell bodies of cells of interest. **d** Schematic view of cells imaged in **e**, **f**. Wild-type embryos were transplanted with cells from a donor embryo expressing Utrophin-GFP (actin probe) and mbCherry. **e** Anterior OP cells converging along the brain surface exhibit canonical morphologies of actively migrating cells, with dynamic filopodia (*white arrowheads*) and actin accumulation at their leading edge. *Green arrows* indicate protrusions that do not belong to the two cells of interest, but to a more anterior cell following them. **f** In cells moving laterally, filopodia and actin are observed at the tip of axons, but not at the level of the cell bodies. *Scale bars*: 10 μm

expression of a dominant-negative (DN) form of the small Rho GTPase RhoA, an upstream regulator of Rock and myosin II[33] (Fig. 5a). There were less transplanted cells in OPs in the DN-RhoA condition than in control transplants at 24 s (control: $n = 18.8$ transplanted cells/placode $\pm 2.0$; DN-RhoA: $n = 7.2 \pm 0.7$), suggesting that many DN-RhoA+ cells do not proliferate, die or are extruded from the embryo during development. We analysed the spatial distribution of living transplanted cells (Fig. 5b) within OPs at the end of morphogenesis (24 s). Along the AP axis, the distribution of DN-RhoA+ cells was more spread than that of control cells, with ectopic cells observed in aberrant anterior and posterior positions (Fig. 5c), indicating that RhoA acts at least partially in a cell-autonomous manner in AP convergence movements. However, the spatial distribution of DN-RhoA+ cells was not affected along the ML axis (Fig. 5c), suggesting that RhoA is not intrinsically required for lateral movements. Consistent with the observed intrinsic role of RhoA in convergence, dynamic accumulations of myosin II-GFP in the front or rear of somata coincided with convergence movements (Fig. 6a, b, Supplementary Movie 7 and Supplementary Fig. 9), suggesting that cell-autonomous myosin II contraction promotes these movements by pulling and pushing[34] the cell bodies forward. By contrast, when cells moved away from the brain and

elongated their axon (lateral movements), myosin II-GFP could not be detected in their cell bodies but rather at the extremity of axons (Fig. 6a, c and Supplementary Movie 8), further supporting the idea that lateral movements of cell bodies do not depend on the RhoA/myosin II pathway.

The phenotype observed upon mosaic perturbation of RhoA (Fig. 5c) suggests that other, RhoA/myosin II-independent mechanisms are at play during convergence. We used a similar transplantation approach to test the involvement of Rac, another RhoGTPase known to regulate actin polymerisation[33], in OP cell movements. We found more DN-Rac+ transplanted cells within OPs than in the control condition (control: $n = 11.9$ transplanted cells/placode $\pm 1.2$; DN-Rac: $n = 17.1 \pm 1.5$), indicating that DN-Rac+ cells survive properly during development. As for RhoA, DN-Rac cells were more scattered than control cells along the AP axis (Fig. 5d), indicating a cell-autonomous requirement for Rac in convergence. Surprisingly, DN-Rac+ cells dispersed more than control cells along the ML axis (Fig. 5d), suggesting that Rac, instead of promoting lateral movements, prevents them, possibly by regulating cell adhesion[35]. These results were reinforced by live imaging of transplanted DN-Rac+ cells (Supplementary Movie 9 and Supplementary Fig. 10). In agreement with a cell-autonomous function of Rac in

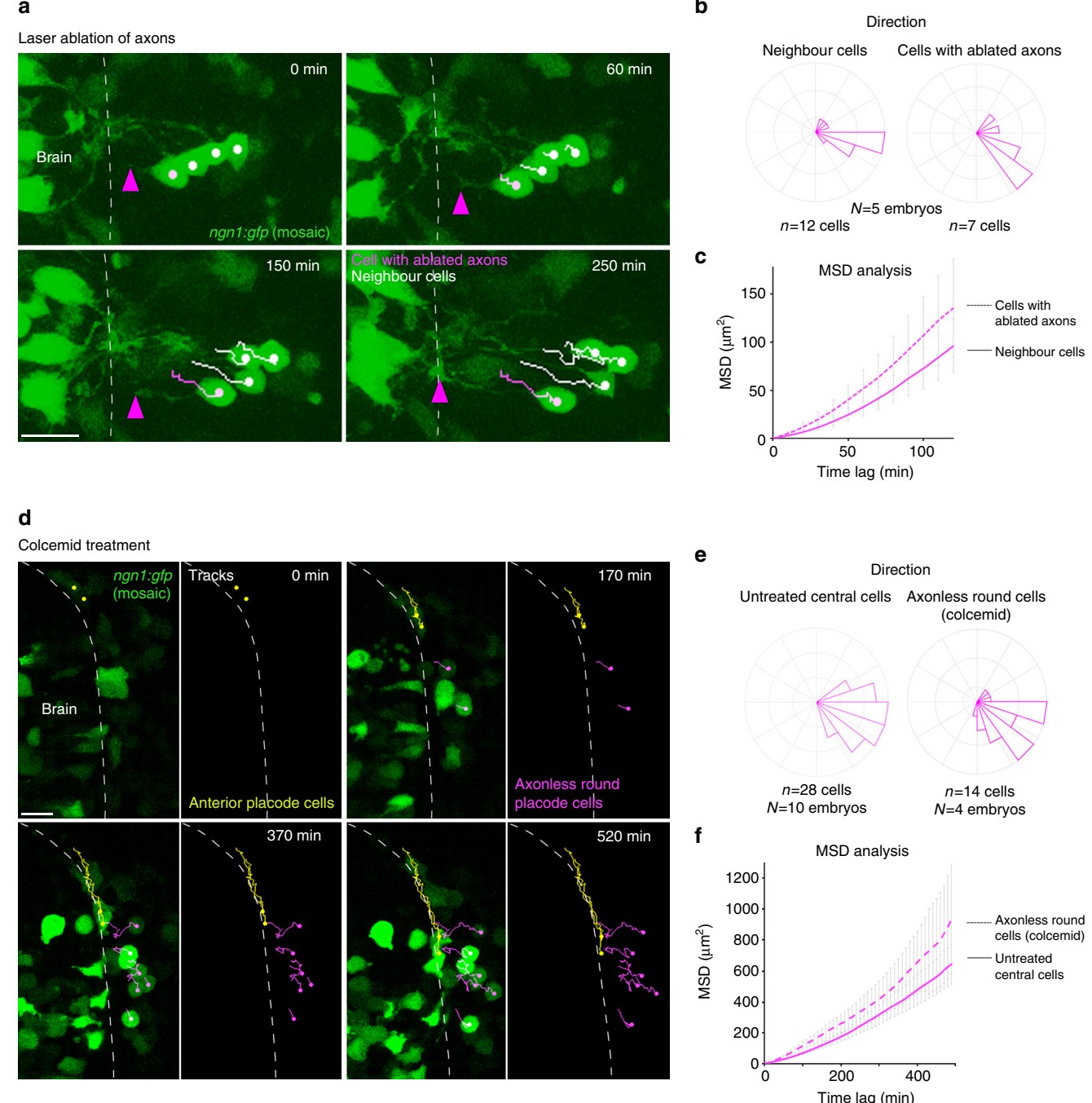

**Fig. 7** Analysis of the role of axons in lateral cell movement. **a** Images extracted from a movie performed on an embryo expressing mosaic *ngn1:gfp* after laser ablation of an axon. Cell tracks are represented in *magenta* for the cell with the ablated axon and in *white* for three neighbouring cells. *Magenta arrowheads* point to the tip of the ablated axon during its regrowth towards the brain surface. **b**, **c** Direction of movements and MSD analysis for the cells with ablated axons as compared with neighbouring cells, during the period of time that preceeds the formation of the new axon/brain contact. **d** Images extracted from a movie performed on an embryo expressing mosaic *ngn1:gfp* and treated with colcemid from 12 s onwards. *Yellow tracks* show the convergence of two anterior cells, and *magenta tracks* show the lateral movements of round, axonless cells. **e**, **f** Direction of movements and MSD analysis for lateral movements of the round axonless central cells, as compared with wild-type untreated central cells. *Scale bars*: 20 μm

convergence, actin-rich filopodia, a hallmark of cells undergoing active migration, were seen in the leading edge of AP-converging cells (Fig. 6d, e and Supplementary Movie 10). By contrast, when cells underwent lateral movements/retrograde axon extension, no obvious actin-rich protrusion could be detected at the level of their cell bodies. Instead, actin was enriched at the extremity of axons near the brain wall, where many active filopodial protrusions could be observed (Fig. 6d, f and Supplementary

Movie 11). Altogether, these findings strongly suggest that convergence movements, but not lateral movements, represent an active cell migration process that depends on the RhoA/ myosin II and Rac/actin machineries.

**The axons are not required for lateral cell body movements.** The axons or their anchoring to the brain could be important for the lateral movement of OP cell bodies. To test the role of axon

anchoring, we laser-ablated individual axons in embryos expressing mosaic *ngn1:gfp* and analysed the effect on lateral movement of the cell bodies. The axon often regrew after ablation, but cell somata exhibited clear lateral movement before the regrown axon contacted the brain (Fig. 7a and Supplementary Movie 12).

During this period of time, the cells with ablated axons moved laterally with similar MSD and direction as their non-ablated neighbours (Fig. 7b, c). These results demonstrate that the attachement of axons to the brain is not necessary for lateral movements of OP cell bodies.

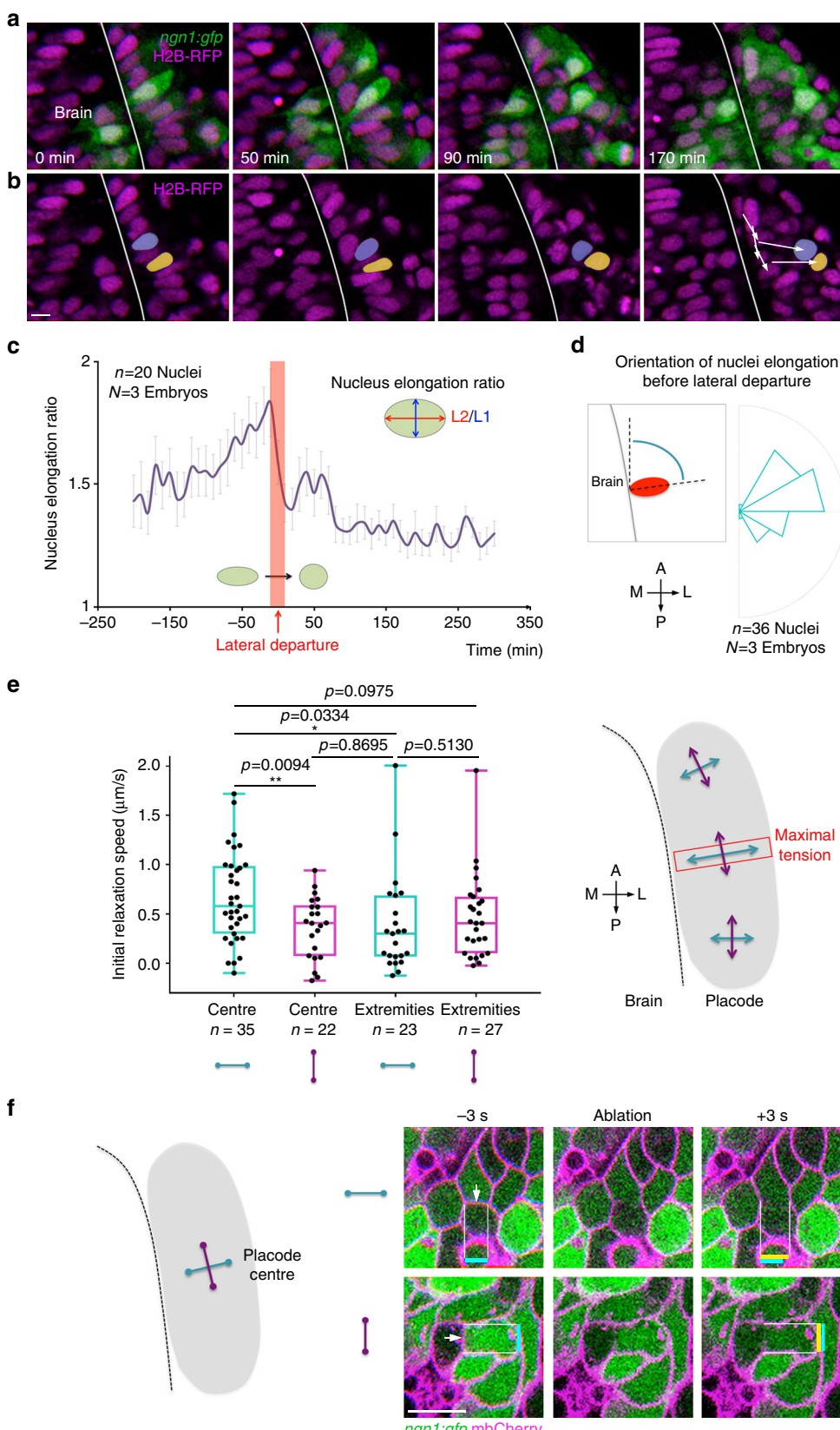

To examine the role of the axon itself in lateral movement, we took advantage of the colcemid treatment condition. In colcemid-treated embryos expressing mosaic *ngn1:gfp*, a significant proportion of GFP + cells exhibited short axons that did not contact the brain, or even no axonal protrusion at all at 24 s (Supplementary Fig. 7d, e). Live imaging and cell tracking in this condition confirmed that cells from the AP extremities converge normally (Fig. 7d and Supplementary Movie 13, yellow tracks), as initially suggested by the absence of placode shape phenotype at 24 s in fixed embryos (Fig. 4a, b). Strikingly, we observed many cells getting round in the centre of the placode. Those round axonless cells moved laterally with similar direction and MSD as compared with untreated central cells (Fig. 7e, f and Supplementary Movie 13, magenta tracks). Thus, microtubules and the axons are not required for the lateral movement of OP cell bodies. Since lateral movements do not depend on intrinsic actomyosin either, they must rather represent a passive, non-autonomous process.

**Mechanical stress is anisotropic in the developing OP**. We hypothesised that lateral cell movements are triggered by extrinsic mechanical forces. Nuclear morphology can be used as readout for mechanical stress: whereas cells undergoing no or isotropic stress have spherical nuclei, cells harbour deformed ellipsoid nuclei in response to anisotropic mechanical stress[36–39]. In agreement with an implication of forces in OP morphogenesis, we observed highly deformed nuclei in the centre of the placode, close to the brain surface; these nuclei were elongated along the ML axis (Fig. 8a, b, left panels, and Fig. 8d). When cells in the close environment of these elongated nuclei were killed with two-photon laser ablation, elongated nuclei immediately got rounder in 20 cases out of 26 (Supplementary Movie 14), suggesting that they undergo anisotropic mechanical stress coming from surrounding cells or tissues. Strikingly, in normal (non-ablated) conditions, elongated nuclei retrieved a round morphology as they started to move away from the brain surface (Fig. 8a–c and Supplementary Movies 15 and 16), suggesting that lateral departure coincides with the relaxation of mechanical stress.

In order to confirm these results and further map the mechanical forces, we probed the distribution of cell–cell tension during placode morphogenesis. To do so, we used two-photon laser ablation to sever individual cell/cell interfaces and measured the initial relaxation speed of the vertices after ablation[27, 40–42]. Tension was measured at 16 s in placode extremities and in the centre, along intercellular contacts oriented parallel or perpendicular to the brain surface (Fig. 8e). The highest tension was measured in the OP centre, along cell/cell interfaces that are perpendicular to the brain (Fig. 8e, f and Supplementary Movie 17). On the basis of our live imaging analysis of cell movements, we hypothesise that this tension anisotropy results from AP compression forces exerted by cells from the edges of the OP tissue. Consistent with this idea, in our movies, cell bodies of OP cells starting their lateral movements appeared to be displaced by direct neighbouring cells crawling towards the placode centre (Fig. 8b, right panel, and Supplementary Movies 15 and 16). Collectively, our data support a scenario in which placodal cells actively migrating from anterior and posterior extremities towards the placode centre exert uniaxial compression forces on central cells, thereby squeezing them away from the brain and contributing to the elongation of their axons.

## Discussion

In the developing nervous system, newborn neurons travel in complex and changing environments to reach their final position, while they grow axons and dendrites to establish functional contacts. In this paper we study the coordination of these processes during the morphogenesis of a sensory organ, the zebrafish OP. We show that the OP forms by a combination of active convergence movements along the brain (Fig. 9a, red) and passive lateral cell displacements (Fig. 9a, blue) during which cell bodies move away from the tip of their axons attached to the brain wall (retrograde axon extension).

What is the origin of the mechanical forces involved in lateral movement? Cell nucleus deformation patterns and laser ablation suggest that cells in the placode centre undergo anisotropic mechanical stress: nuclei are elongated along the ML axis and tension is higher in the same orientation. Anisotropic mechanical stress is unlikely to result from cell-autonomous actomyosin contraction at the cortex, since no obvious cortical actin or myosin II enrichment could be detected in cell bodies in the centre of the placode. Instead, our findings strongly suggest that ML tension and nuclei elongation are due to extrinsic mechanical forces exerted on central cells: either AP compression forces or ML pulling forces or a combination of both. Our analysis of cell movements supports a scenario in which cells actively migrating from the anterior and posterior edges of the placode tissue apply compression on central cells that substantially contributes to their lateral expulsion (Fig. 9b, red).

What drives active convergence movements of OP cells towards the placode centre? We observed two types of cell morphologies during the convergence phase: most often drop-like cells with short leading protrusions contacting the brain, but also, although less frequently, cells with larger lamellipodia-like protrusions (Supplementary Fig. 11), suggesting that different modes of migration coexist during convergence. Live observation of actomyosin dynamics strongly suggest that myosin II contraction and actin polymerisation are both involved in convergence movements. The results of our mosaic perturbation of RhoA and Rac further reinforce the notion that cytoskeleton dynamics acts in a cell-autonomous manner in this process. Cxcl12a/Cxcr4b signalling, which has been shown to be required for OP coalescence[8], is a good candidate for acting as an upstream extracellular signal controlling the activity of RhoA/myosin II and Rac/actin machineries in converging OP cells.

**Fig. 8** Nuclei deformation patterns and mapping of mechanical tension with laser ablation of cell/cell contacts. **a, b** Images extracted from a movie performed on a *ngn1:gfp* embryo injected with H2B-RFP mRNA. In the centre of the placode, the nuclei of cells are initially highly deformed and elongated along the ML axis. In **b**, the nuclei of two cells undergoing lateral movements, drawn in *yellow* and *purple*, retrieve a round morphology right after their lateral departure. **c** Quantification of changes in nuclei elongation ratio before, during and after the lateral departure of OP cells, showing that lateral departure coincides with a decrease in the elongation ratio (averaged on *n* = 20 nuclei from *N* = 3 embryos). **d** Rose plot showing the angle of nuclei elongation right before their lateral departure (*n* = 36 nuclei from *N* = 3 embryos). **e** Laser ablation of cell/cell contacts was performed around 16 s, in OP extremities and in the placode centre, on interfaces oriented parallel (*purple*) or perpendicular (*blue*) to the brain surface. *Graphs* show the initial relaxation speed, used as a proxy for the interface tension, in different locations and orientations. The highest tension was measured in the OP centre, along intercellular contacts that are perpendicular to the brain. *n* indicates the number of ablated cell/cell interfaces in each condition (data pooled from seven experiments). The schematic view on the right summarises the results. *p* values: two-tailed unpaired *t*-test. **f** Representative examples of ablation of cell/cell contacts in the OP centre, oriented perpendicular and parallel to the brain surface. The *white arrows* indicate the ablated cell/cell contacts. *Coloured bars* show the vertex–vertex distance before (*blue*) and right after (*yellow*) ablation. Scale bars: 10 μm

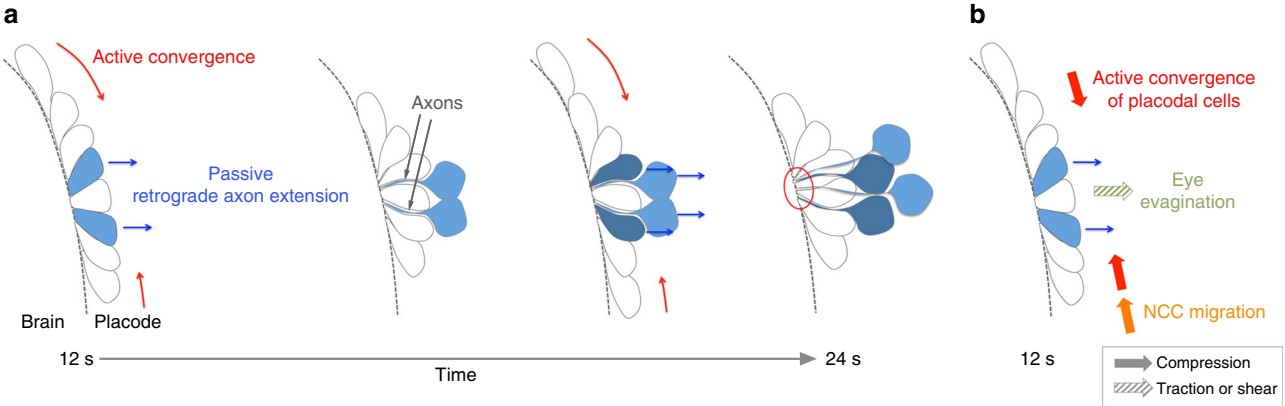

**Fig. 9** Proposed model for the construction of the olfactory circuit during OP morphogenesis. **a** Cells from OP extremities converge towards the centre through active migration along the brain surface (*red*), while cell bodies of central cells passively move away from the brain (*blue*). As they move laterally, central neurons keep contact with the brain surface through long cytoplasmic protrusions, thereby initiating the elongation of their axons (*blue*). Axons thus extend through movements of cell bodies away from static axon tips. We refer to this non-canonical mode of axon elongation as retrograde axon extension. The *red circle* indicates the future entry point of axons in the brain. **b** Possible mechanical forces driving passive retrograde axon extension in the olfactory circuit: compression exerted by actively converging cells from placode extremities (*red*), pushing forces from NCC migration (*orange*) or traction or shear forces exerted by eye evagination movements (*green*)

It has recently been shown that germ band extension in *Drosophila* is partially mediated by tissue-scale pulling forces generated by the adjacent invaginating gut[43, 44]. Do neighbouring tissues also exert mechanical forces on OP cells that participate in their lateral movement? Brain or skin tissues are unlikely to influence lateral cell displacements in the OP, since their movements are either directed anteriorly or are local and diffusive, respectively. During OP morphogenesis, posterior OP cells are followed by cranial NCCs that undergo anteriorward migration and progressively surround the placode between 12 and 20 s[21, 45]. Cranial NCC migration could transmit pushing forces on posterior OP cells and thus participate in the compression of central cells (Fig. 9b, orange). Optic vesicle evagination is concomitant with OP morphogenesis and occurs underneath the OP through lateral tissue flows[46, 47] that could exert shear forces on overlying OP cells, thus contributing to their lateral movements (Fig. 9b, green). Additional experiments are required to clarify the implication of eye and NCC movement in lateral displacement of OP cells.

Our results support a scenario in which retrograde axon extension is driven by extrinsic mechanical forces that either push or pull the cell bodies away from their axon tips attached to the brain surface. In this situation, the lateral displacement of cell bodies is a passive process. Axons anchored to the brain must initially elongate through pure stretching due to forced cell body lateral movements, but after initial stretching of the protrusion novel material (membrane, microtubules) must be added to the axon shaft to accommodate growth. This active process of new material addition likely participates in the retrograde elongation of the axons, in combination with the passive displacement of cell bodies.

The emergence and growth of an axon towards its target is the first step of neuronal polarisation. In textbooks, axon elongation is seen as a growth-cone-driven process, in which the axon extremity moves progressively further away from a cell body[30, 31]. In contrast to this paradigm, we describe here a different mode of axon extension, in which the axon grows by passive retrograde movement of the cell body away from the axonal distal extremity (Fig. 9a, blue cells). This wiring strategy spares the difficulty for the axon to travel through a complex environment and find the brain surface, its intermediate target. How common is this mechanism? A similar retrograde mode of extension has been described for dendrites of *C. elegans* amphid neurons. Strikingly,

dendrite extension occurs in a context of extensive movements of the *C. elegans* embryo[48], raising the interesting possibility that external mechanical forces could also play a role in this process. Other instances of movements of neuronal cell bodies away from the axon tip have been described in the literature, both in the periphery[15] and in the brain, in particular for hindbrain motoneurons and cerebellar granule neurons[49–53], although the underlying dynamics and driving forces remain elusive. Axonal elongation with fixed distal extremity also occurs after the growth cone has reached its final target, and this process has been proposed to depend on mechanical forces imposed by tissue growth[54–56]. Thus, our study of retrograde axon extension and of extrinsic mechanical cues as a driving force calls for analysis of this phenomenon in other developing neuronal circuits.

## Methods

**Fish strains**. Wild-type and transgenic zebrafish embryos were obtained by natural spawning. To obtain the 12 s stage, embryos were collected at 10 am, incubated for 2 h at 28 °C before being placed overnight in a 24 °C incubator. In the text, the developmental times in hpf indicate post-fertilisation hours at 28 °C. Twelve somites correspond to 15 hpf and twenty-four somites to 21 hpf. The OP was visualised using the Tg(8.4neurog1:gfp) line[23], referred to as *ngn1:gfp* in the text. The Tg(actb1:myl12.1-eGFP) line, referred to as *βactin:myosinII-GFP* in the manuscript, was used to visualise myosin II dynamics[57]. All our experiments were made in agreement with the european Directive 210/63/EU on the protection of animals used for scientific purposes, and the french application decree 'Décret 2013-118'. The projects of our group have been approved by our local ethical committee 'Comité d'éthique Charles Darwin'. The authorisation number is 2015051912122771 v7 (APAFIS#957). The fish facility has been approved by the French 'Service for animal protection and health', with the approval number A-75-05-25.

**mRNA and DNA injection**. mRNAs were synthesised from linearised pCS2 vectors using the mMESSAGE mMACHINE SP6 transcription kit (Ambion). The following amounts of mRNA were injected into one-cell stage embryos: 80 pg for H2B-RFP and H2A-CFP[58], 100 pg for mbCherry (membrane Cherry)[58], 100 pg for Kaede, 30 pg for Utrophin-GFP and Doublecortin–GFP, 30 pg for Kif5c560-YFP[28, 29], 200 pg for DN-RhoA (RhoAN19)[59] and 40 pg of DN-Rac (Rac1N17)[59]. pCS2-Kaede, pCS2-Doublecortin–GFP and pCS2-Utrophin-GFP were kind gifts from David Wilkinson, Marina Mione and Marie-Emilie Terret, respectively. To obtain transient transgenic embryos mosaically expressing mbCherry or the *ngn1:gfp* transgene, 40 pg of linearised pCS2-mbCherry[58] and 8.4neurog1:gfp plasmid DNA[23] were injected in one-cell stage embryos, respectively.

**Drug treatments**. Embryos with opened chorion were incubated from 12 to 24 s in four-well plates with the drugs or the equivalent % of DMSO diluted in E3 medium. To block proliferation, embryos were treated with 20 mM hydroxyurea and 150 μM aphidicolin (HUA). To block apoptosis, embryos were incubated in 100

μM of the pan-caspase inhibitor Q-VD-OPh (referred to as caspase inhibitor in the text). Colcemid was used at 100 μM, Blebbistatin at 50 μM and Rockout at 50 μM. Blebbistatin and Rockout treatments did not impair embryo development or viability (Supplementary Fig. 12).

**Immunostaining.** For immunostaining, embryos were fixed in 4% paraformaldehyde, blocked in 5% goat serum, 1% bovine serum albumin and 0.3% triton in PBS for 3 h at room temperature and incubated overnight at 4 °C with primary and secondary antibodies. The following primary antibodies were used: Dlx3b (mouse, 1/500, Zebrafish International Resource Center at the University of Oregon)[21, 22], HuC/D (mouse, 1/200, 16A11 clone, Molecular Probes), acetylated tubulin (mouse, 1/500, 6-11B-1 clone, T6793, Sigma), Zns2 (mouse, 1/500, Developmental Studies Hybridoma Bank)[17], Phospho Histone-H3 (rabbit, 1/200, 06-570, Millipore) and activated Caspase-3 (rabbit, 1/200, AF835, R and D Systems).

**Cell transplantation.** Cells were transplanted from sphere stage donors into wild-type or *ngn1:gfp* host embryos at 50% epiboly or shield stage, targeting the animal pole of the embryo, which gives rise to the OPs and telencephalon[60].

**Live imaging.** Embryos were dechorionated manually and mounted in 0.5% low-melting agarose in E3 medium, in order to obtain a dorsal view of the head (Fig. 1a, b, the orientation of the objective is indicated by a grey arrow). Movies were recorded at the temperature of the imaging facility room (22 °C) on a Leica TCS SP5 AOBS upright confocal microscope or a Leica TCS SP5 MPII upright multi-photon microscope using ×25 (numerical aperture (NA) 0.95) or ×63 (NA 0.9) water lenses. The $\Delta t$ between each frame was 10 min for all our live imaging analysis of cell movements. At 22 °C, it takes 500–600 min for zebrafish embryos to develop from 12 to 24 s stages.

**Kaede photoconversion.** Embryos were injected with Kaede mRNA and mounted as described above for live imaging. Photoactivation was performed using 30 successive scans with the 405 nm laser on a Leica TCS SP5 AOBS upright confocal microscope.

**Laser ablation.** For ablation of cell/cell contacts, ablations were performed in OPs of 14–18 s *ngn1:gfp* embryos injected with mbCherry mRNA to label the membranes. Embryos were mounted in 0.5% low-melting agarose in 1× E3 medium in Ibidi dishes (81158) and imaged using an inverted laser-scanning microscope (LSM 880 NLO, Carl Zeiss) equipped with a ×63 oil objective (1.4 DICII PL APO, Zeiss). Intercellular interfaces were individually severed using a Ti:Sapphire laser (Mai Tai, DeepSee, SpectraPhyics) at 790 nm with <100 fs pulses, using a 80 MHz repetition rate. The two-photon laser was used at 100% power and the number of iterations (between 5 and 10) was chosen to sever cell/cell contacts without creating cavitation[61]. The tension of the cell interface prior to ablation and the speed of opening of the vertices immediately after ablation (initial relaxation velocity) were considered to be proportional[27, 40–42]. For the analysis of laser ablation experiments, the mbCherry signal was first denoised using the PureDenoise ImageJ plugin to improve the accuracy of vertex localisation. To determine the initial relaxation velocity, the vertex–vertex distances of the pre-cut and post-cut interface (three frames after ablation, typically over the first seconds after ablation) were manually measured using ImageJ in a blind procedure. Sometimes we measured a significant decrease in the vertex–vertex distance after the ablation, likely due to the change of focus of the cut interface. These negative velocities thus could not be used in our analysis. A cell/cell contact was considered to be in the OP centre region if located <20 μm from the exact AP centre of the tissue. For ablation of cells, ablations were performed in OPs of 14–18 s *ngn1:gfp* embryos injected with H2B-RFP mRNA to label the nuclei. Ablation of axons were performed in OPs of 20 s wild-type embryos injected with *ngn1:gfp* DNA to obtain mosaic GFP expression. The conditions of ablation (microscope, laser power, frequence) for cells and axons were similar to those used for severing cell/cell contacts.

**Image analysis.** Individual cells were tracked in 3D using the Manual Tracking plugin in ImageJ. The orientation of the movement represents the angle between the cell track and the AP axis, as shown in Supplementary Fig. 4c. 3D MSD analysis was performed with the MSD analyser tool in MATLAB[62]. Plots representing cell tracks merged at their origin were produced with the DiPer program[63]. Rose plots and 3D cell trajectories were generated in MATLAB. 3D counting of nuclei and 3D cell reconstruction analysis were achieved in ImageJ with the 3D ImageJ suite[64] and the 3D Object Counter plugin[65]. Analysis of the correlation between tracks was performed using MATLAB. For each pair of tracks we computed the two-dimensional correlation coefficient as follows:

$$R\left(\vec{r_1}, \vec{r_2}\right) = \frac{\sum_{i=1}^{n}\left(\left(x_i^{(1)} - \overline{x^{(1)}}\right)\left(x_i^{(2)} - \overline{x^{(2)}}\right) + \left(y_i^{(1)} - \overline{y^{(1)}}\right)\left(y_i^{(2)} - \overline{y^{(2)}}\right)\right)}{\sqrt{\sum_{i=1}^{n}\left(\left(x_i^{(1)} - \overline{x^{(1)}}\right)^2 + \left(y_i^{(1)} - \overline{y^{(1)}}\right)^2\right)}\sqrt{\sum_{i=1}^{n}\left(\left(x_i^{(2)} - \overline{x^{(2)}}\right)^2 + \left(y_i^{(2)} - \overline{y^{(2)}}\right)^2\right)}}$$

$R$ is equal to +1 when $\vec{r_1}$ and $\vec{r_2}$ are perfectly correlated, 0 when they are not and −1 when they are perfectly anticorrelated.

**Statistical analysis.** Graphs show mean ± s.e.m. (standard error of the mean), or box and whiskers overlayed with all individual data points. The box and whisker plots were generated with the GraphPad Prism software. $p$ values correspond to two-tailed Student's $t$-test for all figures, except for Fig. 5c, d and Supplementary Fig. 3a where a $\chi^2$-test analysis was carried out (*$p < 0.05$, **$p < 0.01$, ***$p < 0.001$). For $t$-tests exact $p$ values are indicated in the figures. Welch's correction was applied when standard deviations were not equal. We did not check for normality before performing parametric tests ($t$-tests), but a non-parametric Mann–Whitney test was carried out in parallel on all data sets and led to the same conclusions than $t$-tests. No statistical method was used to estimate sample size and no randomisation was performed. Blinding was performed for the analysis of laser ablation experiments (see Laser ablation section in the Methods).

**Data availability.** All relevant data are available from the authors on demand.

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

## Acknowledgements

This work was funded by the Agence Nationale pour la Recherche (ANR blanc project 11-BSV2-0006 to S.S.M.), the Fondation pour la Recherche Médicale (Equipe FRM DEQ20140329544 to S.S.M.), the 'Association pour la Recherche sur le Cancer' (ARC, PJA 20131200051 grant to M.A.B.), the Centre National pour la Recherche Scientifique (CNRS 'Défi mécanobiologie' grant to M.A.B.) and Sorbonne Universités (Emergence 2016 grant to M.A.B., SU-16-R-EMR-11). M.A.B. was supported by a postdoctoral fellowship from the FRM. I.B. belongs to the CNRS research consortium (GdR) 'CellTiss'. We also thank PICT-IBiSA of the UMR3215 and the imaging platform of the IBPS for assistance with microscopy, the IBPS aquatic platform for fish care, P. Marcq and F. Brochart for discussions, and M. Catala, F. Graner, M. Labouesse, F. Robin and C. Vesque for critical reading of the manuscript.

## Author contributions

M.A.B. designed the project. M.A.B., I.B. and S.S.-M. conceived the experiments. M.A.B., I.B., J.S., J.X., S.D.C. and S.S.-M. conducted the experiments and analysed the data. M.A.B. and S.S.M. wrote the paper.

## Additional information

**Competing interests:** The authors declare no competing financial interests.

