## [Peer Review File · Nature Communications]

Reviewers' comments:

Reviewer #1 (Remarks to the Author):

Breau and colleagues perform in this manuscript an extensive description of the morphogenesis of the olfactory placode. The authors use live imaging and cytoskeletal perturbations to get insight on the mechanisms driving convergence of the OP cells and following axonal extension. Looking at membrane, microtubules and nuclei, they observe that axonal extension is the consequence of the cell body displacement, which is unusual and not properly describe so far. They show that convergent cell movement is not affected by microtubule depolymerization and dependent on myosin 2 activity. They also observe a mild phenotype while cells express a dominant negative form of RhoA in mosaic. On the contrary, axonal extension seems not affected by cytoskeletal perturbation and appears to be the consequence of extrinsic forces. Laser dissection showing anisotropy of tension in the central region supports this scenario. Overall, Breau and colleagues developed an interesting study describing an innovative mechanism of axonal extension. The data presented are clear and support some of the conclusions of the manuscript. However, I have some concerns on the extend of some evidences to support the mechanisms proposed by the authors, particularly on the conclusions on the convergent migration.

Specific comments:

1-In the fig 1-d, the tracks of the brain and placode seem correlated, while the authors state: "the movements of OP cells do not follow those of cells from surrounding skins and brain tissues...". The data presented do not support this statement and I would encourage the authors to show more clearly the absence of correlation between the convergence movement of the OP cells and the brain movement or to change the statement in the text that seems contradictory.

2-In the figure 4, the pharmacological perturbations indicate a clear role of myosin 2 contractility in the process. It is however unclear that this is specific to the OP cells. A track analysis, as in fig 1, performed on the perturbations by blebbistatin, rockout and DN rhoA expressed in mosaic would help to disentangle the contribution of the OP cells to the overall movements from the contribution of the surrounding tissues.

3-The authors propose that OP cells actively migrate to generate this convergence movement. This conclusion is based on the observation of myosin foci and filopodia in the OP cells during the convergence movement. As it is, these two observations are insufficient to reach such conclusion as myosin 2 and filopodia are not exclusively involved in cell migration. Correlations between myosin dynamics and individual cell movement, or quantification of polarized protrusions associated to cell shape changes could be performed to reinforce the conclusions. Affecting cell adhesion molecules, other rhoGTPases such as Rac or chemokines signaling could also bring additional strong support to their model.

4-The authors perform laser dissection of cell-cell junction to estimate anisotropy of tension in the central region. From their measurements they conclude that this anisotropy arises from compressive forces. It is not clear whether this anisotropy of tension reflects compressive forces or simply originates from cell polarity. Doing cuts at several stages of convergence, where compressive forces are likely to be different, would allow to dissect between the two possibilities and strongly improve the conclusions of the manuscript.

5-The description of the retrograde axonal growth the authors are performing is original and nicely characterized by the observation of nuclear deformations. Similar elongation have been observed previously in the case of motor neurons (Wada et al, Development 2005) and should be mentioned.

6-At the end of the first paragraph, the authors state: "Collectively, these experiments rule out a major implication of cell death, cell proliferation and cell size or shape changes in OP morphogenesis." This is not entirely correct as cell shape changes occur during OP morphogenesis (and are described in the following of the manuscript). I would invite the authors to modify their statement.

7-Videos of perturbed cases (for apoptosis, proliferation, colcemid or blebbistatin) are not present to help us evaluate the extent of the perturbation on the dynamics of the process. An additional video combining all the perturbations with WT would be very helpful for the reader to evaluate the extent of the different perturbations.

Reviewer #2 (Remarks to the Author):

Review for paper:

Extrinsic mechanical forces mediate retrograde axon extension in a developing neuronal circuit

Breau MA, Bonnet I, Stoufflet J, Xie J, De Castro S, Schneider-Maunoury S

Summary:

In this study the authors characterize the movements of neurons in the olfactory placode of the developing zebrafish. They show that this displacement correlates with the axons presumably attached at the initial point of neuronal movement and cell bodies moving away, a mode they refer to as retrograde axon extension. This is different from many other neuronal migration modes in which the cell body actively moves and the axon emerges from the trailing process.

While the assortment of data presented in this study has some interesting new aspects, it currently lacks a mechanistic explanation of the phenomena observed that goes beyond correlations. Thus, the authors either have to substantiate their claims with additional experiments or could send this paper to a more specialized journal in the field of developmental biology.

Major Points:

General open questions

- One interesting question that was not asked at all in this study, but would be an important contribution to understanding the observed phenomena, is how axons are anchored to the 'brain surface'? What is the substrate? Are axons linked to ECM and/or components secreted from the surrounding brain cells? Some work on the linker proteins has been done in a similar process of retrograde dendrite extension in *C. elegans*, studies that are also mentioned in the discussion of this work. These could serve as potential entry routes into this question.

- Along the same lines, it is not addressed whether the retrograde extension of the cell body depends on axon anchorage. What happens when for example axons are cut using laser ablation as done for other experiments? Does the process still occur? Vice versa, if cells are retained from moving does the axon still elongate and does it maybe twist around the non-moving cell?

I believe that the role of the axon is an important area to explore that would help to understand the phenomenon observed.

Experimental comments

- The experiments in which the authors apply colcemid (side note: it should be colcemid everywhere not colcemide) to the embryos need further evaluation. At this point the authors present a 'before and after' analysis of the condition. They state that axons in this condition do not contact the brain any more. However, they do not reveal whether cell movement occurred before the axons were detached or whether cells retracted the axons after they reached their final location. In both cases microtubule might still play a role in cell body displacement. Life imaging and single cell tracking should be added for this condition as well as a genetic mosaic condition as in the case of actomyosin (see below).

- In the experiments in which the authors treated embryos with blebbistatin and Rockout the same problem applies. It is not clear what stage of movement is affected or whether generally the whole embryo stopped developing. This is a possibility as for example the 50 μM blebbistatin used seems like a rather harsh condition. It is possible that all developmental processes in the zebrafish embryo stop at this high drug concentration. Usually a concentration of 20 μM to 25 μM and a short pulse of blebbistatin is used. Their interpretation once more needs to be backed up with live imaging.

- For the involvement of actin and myosin the authors did try to add a genetic condition but the outcome of these experiments is hard to interpret. To me it looks as if all DN-RhoA cells are dead in the pictures chosen to present. This experiment could be better designed. Instead of constitutive expression of RNA, which has an adverse effect on viability of cells too early in development, a better strategy would be to inject an inducible DNA construct, for example a heat shock inducible DN-RhoA. In addition, the authors are in a very good position of having a specific promoter, under which they could express proteins specifically in the tissue of their interest. They should take advantage of this by, e.g. overexpressing the non-phosphorylatable version of myosin II or other genetic conditions that underline their findings in the drug condition.

- What is confusing is that the authors, after having stated that they assume actomyosin dependent forces to play a role in retrograde axon extension, speculate that actin rich filopodia seen at the leading edge play a role in the cell movement. This would be a different mechanism depending more on actin polymerization phenomena, for example via the Arp2/3 complex. What is more is that a protrusion driven mechanism would need a substrate to generate friction force. This should be addressed and tested. It should further be stated how the authors think about the interplay of these different actin dependent migratory modes in their model.

- At this stage the interpretation of nuclear shape changes and the role of mechanics for the lateral cell movements is pure speculation and correlation. Nuclear shape changes could also result from intracellular shape changes or other intracellular reorganisation. If the authors want to keep this claim they need to show that interference with the surrounding tissue indeed has an effect on nuclear shape.

- When the authors apply the laser ablation I do not see a difference between the two panels in the movie presented. I looked at them very hard and many times. I also did a blind test with some of my lab members (without breaking confidentiality of the manuscript) and none of them saw a difference between the two panels. Maybe a more obvious example could be chosen or the analysis could be made more clear.

Overall the influence of tissue wide mechanics in the process of cell movement needs further validation.

Minor points:

- It was a bit annoying that no panels (a,b,c) were stated for Supplementary Figures in the text and did cost some time to find the respective panel referred to, this should be changed.

- In Figure 1 it is hard to imagine how single cells could be tracked in 3D when all surrounding cells are labelled. A representative tracking movie featuring tracked cells from the three different areas should be added.
- What is the time resolution (and thereby the Δt of MSDs) of imaging described in Figure 1? It would be an advantage to also show standard MSD graphs for clarity, not only the log/log plot.
- In Figure 1 the authors state that cells were tracked in 3D but all graph show 2D tracks, this should be explained.
- Figure 2 b,c,d Filapodia are not obvious in the Figures presented. Maybe a zoom in would help. The same applies for Figure 3 a where axon bundles are hard to see.
- In supplementary Figure 3a what kind of mean is shown? Is it an arithmetic mean? How many placodes were analysed? What do the error bars represent?
- The authors should state the number of individual experiments/embryos that were used to generate their specific $n=x$ (for example: $n=34$ cells from 4 embryos/imaging experiments)
- The authors should explain, why the live imaging was performed at a rather low temperature of 22°C instead of the widely used 28°C.
- It would be good to add a scheme or bright field image of the head of a zebrafish embryo at relevant developmental stages as Figure 1A. That would help non-experts to orient themselves.
- The references for the Kif5c560-YFP construct are inconsistent: refs. 28, 29 are cited in the results section while ref. 32 is cited in the materials and methods section. The refs. 28, 29 are the correct ones.
- The combination of green and red in many figures is not suitable for colour-blind readers and also does not give the best possible contrast.
- The bar graphs used for a lot of data representation are an inferior way of presenting the data. A better way to present it, in my opinion, would be a boxplot analysis with all data points overlaid.

Reviewer #3 (Remarks to the Author):

Breau et al. observed in detail dynamic movements of olfactory placode (OP) cells and neighboring cells in zebrafish, and found that OP cells first converge towards the center of the placode along the A-P axis and then move laterally to form a spherical cluster. Perturbing the RhoA/Rock/myosine II pathway in combination with mosaic analysis revealed that convergence movements, but not lateral movements, are active, myosin II-dependent cell migration. Furthermore, by quantitative analysis of nuclei deformation and measuring the initial relaxation speed after laser ablation of cell-cell interfaces, the authors demonstrated that A-P compression forces from actively converging OP cells trigger the lateral cell movements.

In addition to the cellular movements of the developing olfactory placode, the authors observed an interesting mode of axon formation, called retrograde axon extension, where the axon elongation coincides with displacement of cell bodies away from axon tips anchored to the brain surface.

The study is very well executed and relies on a multiplicity of approaches to elucidate dynamics of OP morphogenesis and its underlying mechanisms. The quality of the data and the presentation is very high. This paper presents a major advance towards understanding of how spherical olfactory placodes form from the elongated cellular field.

My major concern with this paper is the interpretation and/or the definition of the retrograde axon extension. The authors state that the retrograde axon extension is "original" in the zebrafish olfactory neurons (line 74-75; line 194-195). It is, however, unclear for me what process/mechanism of the axon extension described here is original. As in the Discussion section, there are literatures showing a similar mode of axon extension in the mammalian brain (reviewed by Hatanaka et al., Proc Jpn Acad Ser B Phys Biol Sci., 2016): e.g., cerebellar granule cells form trailing processes during their radial migration, and the trailing processes give rise to axons. If the authors think that the contribution of "passive" displacement of soma to the retrograde axon extension is the original mechanism, the procedure of axon extension (not completely novel) should be described in distinction from the novelty of underlying mechanism, and the term "original" should be used more carefully.

The authors also state that the retrograde axon extension is a passive process (line 239-240; line 242, subheading; Fig 7). However, I think it is not a simple passive process but a combinatorial process of both active and passive modes. Although the perturbation of microtubule polymerization does not affect lateral movements of cell bodies, axon elongation itself requires microtubule polymerization, an active cellular event, which could be induced concomitantly with the passive, non-autonomous, lateral movement of soma.

Minor points:

Fig. 1, legend: Number of cells tracked should be described (c, for anterior, central and posterior cells, respectively). Summary of the movements should be (f), not (e).

Fig. 5, legend: "et" should read "and".

Supplementary Fig. 4, legend: Number of cells tracked should be described (d, e, and f).

Is mbCherry different from mCherry? Is this a membrane-targeted version? If so, please explain when it appears the first time in the text.

Response to reviewers

We thank the reviewers for their thoughtful comments on the manuscript. As detailed below, we have addressed all of points raised by the reviewers with modifications to the figures and text, and additional data. These have greatly improved the manuscript and substantiate our conclusion that extrinsic mechanical forces mediate the retrograde extension of axons in the zebrafish olfactory circuit.

Reviewers' comments:

Reviewer #1 (Remarks to the Author):

Breau and colleagues perform in this manuscript an extensive description of the morphogenesis of the olfactory placode. The authors use live imaging and cytoskeletal perturbations to get insight on the mechanisms driving convergence of the OP cells and following axonal extension. Looking at membrane, microtubules and nuclei, they observe that axonal extension is the consequence of the cell body displacement, which is unusual and not properly describe so far. They show that convergent cell movement is not affected by microtubule depolymerization and dependent on myosin 2 activity. They also observe a mild phenotype while cells express a dominant negative form of RhoA in mosaic. On the contrary, axonal extension seems not affected by cytoskeletal perturbation and appears to be the consequence of extrinsic forces. Laser dissection showing anisotropy of tension in the central region supports this scenario. Overall, Breau and colleagues developed an interesting study describing an innovative mechanism of axonal extension. The data presented are clear and support some of the conclusions of the manuscript. However, I have some concerns on the extend of some evidences to support the mechanisms proposed by the authors, particularly on the conclusions on the convergent migration.

Specific comments:

1-In the fig 1-d, the tracks of the brain and placode seem correlated, while the authors state: "the movements of OP cells do not follow those of cells from surrounding skins and brain tissues...". The data presented do not support this statement and I would encourage the authors to show more clearly the absence of correlation between the convergence movement of the OP cells and the brain movement or to change the statement in the text that seems contradictory.

To address this point, we analysed the correlation between the tracks of brain cells and those of placode cells in the central and posterior placodal regions (anterior brain cells are negative for *ngn1:gfp* and thus were not tracked). The results show that there is no correlation between brain and placode cells in the central region, and a partial correlation in the posterior region, as compared with the high correlation observed between placode cells in each region. These new data were included in Supplementary Figure 4g, and the text of the results section was modified accordingly, lines 150-153. The method used to calculate the correlation coefficient between two cell tracks is described in the Methods/Image analysis section, lines 487-491.

2-In the figure 4, the pharmacological perturbations indicate a clear role of myosin 2 contractility in the process. It is however unclear that this is specific to the OP cells. A track analysis, as in fig 1, performed on the perturbations by blebbistatin, rockout and DN rhoA expressed in mosaic would help to disentangle the contribution of the OP cells to the overall movements from the contribution of the surrounding tissues.

To address the point raised by the reviewer, we first performed live imaging and tracked placode, skin and brain cells in rockout (N=4 embryos) and in blebbistatin (N=2 embryos) conditions (Supplementary Movie 6). The tracking data are presented in new Supplementary Figure 8 and commented in the Results section lines 213-222. The movements of anterior and central placode cells, but not posterior cells, were affected in both drug conditions. Surprisingly, skin cells were also affected and showed clear anteriorward movement in both drug conditions, which is clearly different from the unoriented diffusive-like behaviour of skin cells in controls. This live imaging analysis brings new information about the phenotype observed in blebbistatin and rockout-treated embryos, but the skin cell forward movements does not allow us to conclude about the placode or cell autonomy of the defects.

To address the question of the cell autonomy, we previously achieved mosaic expression of a dominant negative form of RhoA with transplantation experiments. Analysis of the spatial distribution of transplanted cells at 24s suggested that RhoA acts at least partially in a cell-autonomous manner in convergence movements, but not in lateral movements. We attempted to perform live imaging of DN RhoA-expressing cells but this proved difficult, due to the small numbers of transplanted cells found in the OP in this condition (as initially said in the text, those cells may show decreased proliferation or increased cell death before or during OP morphogenesis).

To further investigate the cell autonomy of OP cell movements, we used the same approach to mosaically perturb Rac, another Rho GTPase known to regulate actin polymerisation. As explained in detail below in the response to Point 3 of the reviewer 1, the results largely reinforce the conclusion of the DN RhoA experiments. The number of DN Rac-expressing cells per placode was not decreased as compared with controls, which allowed us to analyse the behaviour of DN Rac cells in live imaging experiments. This analysis confirmed impaired convergence movements but no overt defect in lateral movements of cells expressing mosaic DN-Rac (Supplementary Video 9 and Supplementary Figure 10). Mosaic DN RhoA and DN Rac perturbation, together with the analysis of actomyosin dynamics, strongly suggest that convergence movements are active (autonomous), whereas lateral movements are passive (non-autonomous). For more details see the response to Point 3 of the reviewer 1 below.

3-The authors propose that OP cells actively migrate to generate this convergence movement. This conclusion is based on the observation of myosin foci and filopodia in the OP cells during the convergence movement. As it is, these two observations are insufficient to reach such conclusion as myosin 2 and filopodia are not exclusively involved in cell migration. Correlations between myosin dynamics and individual cell movement, or quantification of polarized protrusions associated to cell shape changes could be performed to reinforce the conclusions. Affecting cell adhesion molecules, other rhoGTPases such as Rac or chemokines signaling could also bring additional strong support to their model.

We performed additional experiments to visualise the dynamics of Myosin II-GFP, by transplanting cells from *βactin:myosinII-GFP* transgenic donors injected with mbCherry mRNA, into WT embryos. We analysed the correlation between Myosin II-GFP dynamic accumulations and the net speed of the cells during anteroposterior migration (data were pooled from n=9 cells from 3 embryos, 543 time points analysed in total). This is presented in Supplementary Figure 9 of the revised manuscript and in the text lines 235-239. The results show that myosin II dynamic accumulations partially correlate with the net movements of anteroposterior migrating cells, which further supports our previous conclusion that intrinsic activity of the RhoA/myosin II machinery promotes convergence movements.

This partial correlation, and the phenotype observed upon mosaic perturbation of RhoA, suggest that other, RhoA/myosin II-independent mechanisms are at play. As suggested by the reviewer, we tested the involvement of Rac, another RhoGTPase mostly known to

regulate actin polymerisation, in convergence and lateral movements. To do so, we transplanted DN Rac-expressing cells into WT embryos and analysed spatial cell distribution at 24s, as done previously for RhoA. The results of DN Rac mosaic experiments largely confirmed those of DN RhoA. As for DN RhoA, the distribution of DN Rac-expressing cells was more spread than that of control cells along the AP axis, with ectopic cells observed in aberrant anterior and posterior positions (new Figure 5d). Surprisingly, DN Rac-expressing cells dispersed more along the ML axis than control cells, suggesting that Rac, instead of promoting lateral movements, prevents them, likely through an effect on cell adhesion (Ratheesh et al., 2013, *Prog Mol Biol Transl Sci.* 116, 49-68). These results were further supported by a live imaging analysis of transplanted DN Rac-expressing cells, showing altered convergence movements but overall unaffected lateral movements, as compared to the behaviour of control transplanted cells (Supplementary Video 9 and Supplementary Figure 10).

Altogether our new results bring additional support to our model in which anterior and posterior OP cells migrate first actively towards the centre of the placode, then move laterally in a passive manner. The RhoA/myosin II and Rac/actin machineries both promote active convergence. The Results paragraph entitled "Active convergence cell movements coordinate with passive lateral soma displacements" has been remodelled to include these additional data (lines 198-263). A paragraph about the mechanisms of convergent cell migration in the placode was also added to the Discussion section, lines 339-349.

4-The authors perform laser dissection of cell-cell junction to estimate anisotropy of tension in the central region. From their measurements they conclude that this anisotropy arises from compressive forces. It is not clear whether this anisotropy of tension reflects compressive forces or simply originates from cell polarity. Doing cuts at several stages of convergence, where compressive forces are likely to be different, would allow to dissect between the two possibilities and strongly improve the conclusions of the manuscript.

To our knowledge, in laser ablation experiments, no direct relationship between the initial speed of vertex retraction and cell polarity has been established so far. In contrast, a clear relationship has been established between the initial speed of vertex retraction and the tension of the contact prior to ablation (Rauzi et al., 2008, *Nat Cell Biol* 10, 1401-10; Rauzi et al., 2010, *Nature* 468, 1110-4; Sugimura et al., 2016, *Development* 143, 186-96). Indeed, when we perform laser ablation of a cell/cell contact, vertices at the extremities of this interface move away from each other: by measuring the initial velocity of this relaxation immediately after severing, we estimate the tension prior ablation, up to an unknown prefactor that depends on the dissipation. This dissipation makes the viscous drag on the vertices' movement. Consequently, when we measure anisotropy in initial relaxation, this anisotropy could arise from anisotropy in tension or anisotropy in the viscous drag. The dissipation originates from the homophilic interactions of cells that could be seen as a viscous interaction. It is commonly considered as homogeneous and constant at the scale of the experiment, and thus anisotropy of velocity retraction comes from anisotropy in tension.

In the first version of the manuscript, we wrote "The highest tension was measured in the OP centre,[...] an anisotropy that could result from AP compression forces exerted by cells from the edges of the OP tissue". The idea that anisotropy in tension is a consequence of compressive forces exerted by cells migrating from the extremities of the placode is proposed as an hypothesis, not as a conclusion. To make it clearer in the text, we changed the sentence as follows : "The highest tension was measured in the OP centre, along cell/cell interfaces that are perpendicular to the brain [...]. Based on our live imaging analysis of cell movements, we hypothesise that this tension anisotropy results from AP compression forces exerted by cells from the edges of the OP tissue" (lines 308-311).

5-The description of the retrograde axonal growth the authors are performing is original and nicely characterized by the observation of nuclear deformations. Similar elongation have been observed previously in the case of motor neurons (Wada et al, Development 2005) and should be mentioned.

Other cases of retrograde axon extension were already cited in the Discussion section, including hindbrain motor neurons. We now specifically comment the case of hindbrain motor neurons and we added the reference Wada et. al., Development 2005, lines 381-384.

6-At the end of the first paragraph, the authors state: "Collectively, these experiments rule out a major implication of cell death, cell proliferation and cell size or shape changes in OP morphogenesis." This is not entirely correct as cell shape changes occur during OP morphogenesis (and are described in the following of the manuscript). I would invite the authors to modify their statement.

We agree with the reviewer that this was an overstatement. What we show is that the reduction in cell size associated with cell division is not involved in changes in the shape of the OP tissue. We corrected the title of the paragraph and the sentence accordingly, lines 114,115.

7-Videos of perturbed cases (for apoptosis, proliferation, colcemid or blebbistatin) are not present to help us evaluate the extent of the perturbation on the dynamics of the process. An additional video combining all the perturbations with WT would be very helpful for the reader to evaluate the extent of the different perturbations.

In addition to new movies in the blebbistatin and knockout conditions mentioned in response to Point 2 (Supplementary Video 6), we also performed new movies in embryos treated with inhibitors of proliferation (HUA, Supplementary Video 3) and colcemid (Supplementary Video 13 and Figure 7). We did not perform any movie on Caspase inhibitor-treated embryos since we had no evidence for a role of cell death in OP morphogenesis or cell number.

Reviewer #2 (Remarks to the Author):

Review for paper:

Extrinsic mechanical forces mediate retrograde axon extension in a developing neuronal circuit

Breau MA, Bonnet I, Stoufflet J, Xie J, De Castro S, Schneider-Maunoury S

Summary:

In this study the authors characterize the movements of neurons in the olfactory placode of the developing zebrafish. They show that this displacement correlates with the axons presumably attached at the initial point of neuronal movement and cell bodies moving away, a mode they refer to as retrograde axon extension. This is different from many other neuronal migration modes in which the cell body actively moves and the axon emerges from the trailing process.

While the assortment of data presented in this study has some interesting new aspects, it currently lacks a mechanistic explanation of the phenomena observed that goes beyond correlations. Thus, the authors either have to substantiate their claims with additional experiments or could send this paper to a more specialized journal in the field of developmental biology.

Major Points:

General open questions

*- One interesting question that was not asked at all in this study, but would be an important contribution to understanding the observed phenomena, is how axons are anchored to the 'brain surface'? What is the substrate? Are axons linked to ECM and/or components secreted from the surrounding brain cells? Some work on the linker proteins has been done in a similar process of retrograde dendrite extension in *C. elegans*, studies that are also mentioned in the discussion of this work. These could serve as potential entry routes into this question.*

[UNPUBLISH DATA REDACTED BY EDITORIAL TEAM UPON AUTHORIAL REQUEST]

We consider that this very interesting question of the mechanisms of axon anchoring requires a completely new project including the functional analysis of several molecular candidates, which in our opinion is beyond the scope of the present work.

- Along the same lines, it is not addressed whether the retrograde extension of the cell body depends on axon anchorage. What happens when for example axons are cut using laser ablation as done for other experiments? Does the process still occur? Vice versa, if cells are retained from moving does the axon still elongate and does it maybe twist around the non-moving cell?

I believe that the role of the axon is an important area to explore that would help to understand the phenomenon observed.

In order to address this important question of whether lateral/retrograde cell body movements depend on axon anchorage, we conducted a series of experiments detailed below, whose results are presented in a new chapter of the Results section, entitled « The axons are not required for lateral movement of cell bodies » (main text lines 265-286, new Figure 7, Supplementary Videos 12 and 13).

1) We laser ablated individual axons during the lateral movement of OP cells in embryos expressing mosaic *ngn1:gfp*, and tracked the movement of the affected cells as well as neighbouring unaffected *ngn1:gfp*⁺ cells. Axons often regrew after ablation but the cell soma exhibited normal lateral movement before the regrown axon contacted the brain. This experiment clearly demonstrates that attachment of the axon to the brain is not required for lateral movement of the cell body.

2) We performed movies on embryos expressing mosaic *ngn1:gfp* and treated with colcemid. Observation of cell morphology revealed that many cells got round in the center of the placode. Those round, axonless cells moved laterally with similar direction and MSD as compared with untreated central cells. This experiment confirms that microtubules are not necessary for lateral movements and shows that not only axon anchoring, but the axon itself, is not required for the lateral displacement of cell bodies within the OP.

In addition, as suggested by the reviewer, we tried several genetic conditions to perturb microtubule polymerisation and/or stability, including overexpression of the microtubule-severing enzymes Katanin and Spastin, but unfortunately none of them perturbed the formation of the axons in our system. In particular, we performed mosaic overexpression of Stathmin, a regulator of microtubule dynamics known to favour microtubule destabilization. We made use of a *hsp70 :stathmin1-Kate2* construct (Icha et al., JCB 2016, 215: 259-275; a kind gift from Dr. Caren Norden). Heat-shock-mediated overexpression of Stathmin1 leads to loss of the basal process and impaired basal translocation in RGCs of the zebrafish retina (Icha et al., JCB 2016). In Figure R2 for the reviewers (see below), we present three examples of OPs with neurons that overexpress Stathmin1-Kate2 (green) and yet have elongated axons (arrowheads in Fig. R2). We conclude that Stathmin overexpression is not a good tool to study the role of axons in our system.

Overall, we think that the data presented in Figure 7 and Supplementary Videos 12 and 13 show compelling evidence that axons or their anchoring to the brain are not required for lateral cell movements during OP morphogenesis.

Figure R2: Stathmin overexpression in isolated OP neurons does not prevent axon formation. Three examples are presented, from three distinct embryos in which mosaic overexpression of Stathmin was achieved. In a, b and c, Stathmin-expressing cells are shown in green and nuclei in magenta. White lines indicate the brain surface. In a', b' and c', only Stathmin staining is shown. Arrowheads point to axons of stathmin-expressing OP neurons. Scale bar is 25 μ m. **Methods:** Embryos were injected with the *hsp70:stathmin1-mKate2* plasmid, heat-shocked at the 10-11s stage for 25 min at 37°C and then returned to 28°C for further development. They were fixed at 24s and stained for DAPI.

Experimental comments

- The experiments in which the authors apply colcemid (side note: it should be colcemid everywhere not colcemide) to the embryos need further evaluation. At this point the authors present a 'before and after' analysis of the condition. They state that axons in this condition do not contact the brain any more. However, they do not reveal whether cell movement occurred before the axons were detached or whether cells retracted the axons after they reached their final location. In both cases microtubule might still play a role in cell body displacement. Live imaging and single cell tracking should be added for this condition as well as a genetic mosaic condition as in the case of actomyosin (see below).

As explained in the response to the reviewer's previous point, live imaging and cell tracking analysis are now presented for colcemid treatment (Figure 7 and Supplementary Video 13), showing that in this condition a lot of cells acquire a round morphology in the center of the placode, close to the brain surface. These round, axonless cells undergo normal lateral movement, demonstrating that microtubules, and the axon itself, are not necessary for lateral displacements of the cell bodies.

- In the experiments in which the authors treated embryos with blebbistatin and Rockout the same problem applies. It is not clear what stage of movement is affected or whether generally the whole embryo stopped developing. This is a possibility as for example the 50 μM blebbistatin used seems like a rather harsh condition. It is possible that all developmental processes in the zebrafish embryo stop at this high drug concentration. Usually a concentration of 20 μM to 25 μM and a short pulse of blebbistatin is used. Their interpretation once more needs to be backed up with live imaging.

Early zebrafish embryos are very sensitive to blebbistatin treatment, Myosin II being required for early cleavage and epiboly (see for instance Chai et al 2015, Biophysical Journal 109: 407-414). However, this is not the case at later stages, and this range of blebbistatin concentrations (50 to 100 μM) has been used for zebrafish embryos in other studies (Ernst et al 2012, Development 139: 4571-4581). In our hands, zebrafish embryos develop correctly when treated with 50 μM blebbistatin or rockout from the 12s stage. This is now illustrated in supplementary Figure 11 and mentioned in the Methods section, lines 424-425. It should also be noted that no massive cell death was observed in live imaging experiments on blebbistatin and rockout-treated embryos (new Supplementary Video 6).

- For the involvement of actin and myosin the authors did try to add a genetic condition but the outcome of these experiments is hard to interpret. To me it looks as if all DN-RhoA cells are dead in the pictures chosen to present. This experiment could be better designed. Instead of constitutive expression of RNA, which has an adverse effect on viability of cells too early in development, a better strategy would be to inject an inducible DNA construct, for example a heat shock inducible DN-RhoA. In addition, the authors are in a very good position of having a specific promoter, under which they could express proteins specifically in the tissue of their interest. They should take advantage of this by, e.g. overexpressing the non-phosphorylatable version of myosin II or other genetic conditions that underline their findings in the drug condition.

Since we observed less DN RhoA-expressing cells than control cells in our transplanted embryos, we hypothesised that DN RhoA expression leads to a reduction in cell survival or cell proliferation. Nevertheless, the DN RhoA cells we analysed at 24s were all alive, as illustrated with DAPI staining (New Figure 5b, insets), and all the cells whose distribution was analysed and quantified in this study were living cells.

In order to overcome this survival or proliferation problem, we attempted to use hsp70:GFP-T2A-DNRhoA, hsp70:GFP-T2A-DNRock2a and hsp70:GFP-T2A-DNRac1 constructs (kind gifts from Felix Loosli, Herder et al., Development 140:2787-97). We injected them in 1-cell stage embryos, heat shocked at 11s and analysed GFP expression at 24s, but very low GFP+ cell numbers were found in these three conditions, in comparison with the hsp70:GFP control construct.

However, our conclusions about the cell-autonomy of the convergence and lateral movements were further validated by the results of mosaic DN Rac expression, which gave similar defects than with DN RhoA in the absence of cell death (see response to Point 3 of Reviewer 1).

- What is confusing is that the authors, after having stated that they assume actomyosin dependent forces to play a role in retrograde axon extension, speculate that actin rich filopodia seen at the leading edge play a role in the cell movement. This would be a different mechanism depending more on actin polymerization phenomena, for example via the Arp2/3 complex. What is more is that a protrusion driven mechanism would need a substrate to generate friction force. This should be addressed and tested. It should further be stated how the authors think about the interplay of these different actin dependent migratory modes in their model.

We agree with the reviewer that the study of OP cell movements was incomplete and biased towards Myosin II-driven mechanisms. We have now added a series of experiments to precise the mechanisms, including mosaic expression of DN Rac and analysis of the correlation between Myosin II dynamic accumulations and convergence cell movements. This is discussed in the response to Point 3 of Reviewer 1's comments. We also added a paragraph in the Discussion section, lines 339-349, to discuss the different modes of migration during convergence.

- At this stage the interpretation of nuclear shape changes and the role of mechanics for the lateral cell movements is pure speculation and correlation. Nuclear shape changes could also result from intracellular shape changes or other intracellular reorganisation. If the authors want to keep this claim they need to show that interference with the surrounding tissue indeed has an effect on nuclear shape.

We previously observed highly deformed nuclei (elongated along the ML axis) in the centre of the placode, close to the brain surface. To test whether the deformation of these nuclei arise from mechanical stress coming from the environment, we used biphoton laser ablation to kill cells in the close vicinity of these elongated nuclei. In 20/26 cases, elongated nuclei immediately changed their shape after ablation to become rounder. Two examples are shown in Supplementary Video 15. This result indicates that cells in the center of the placode undergo anisotropic mechanical stress exerted by surrounding cells or tissues causing the elongation of their nuclei (mentioned in the Results section, lines 296-299).

- When the authors apply the laser ablation I do not see a difference between the two panels in the movie presented. I looked at them very hard and many times. I also did a blind test with some of my lab members (without breaking confidentiality of the manuscript) and none of them saw a difference between the two panels. Maybe a more obvious example could be chosen or the analysis could be made more clear.

We thank the reviewer for pointing this out. The problem did not come from the chosen example but from the time scale of the movie, as explained below.

The tension of the cell/cell contact prior to ablation and the immediate relaxation speed of the vertices are considered to be proportional (Sugimura et al., 2016, *Development* 143, 186-96). To determine the initial relaxation speed in our experiments, the vertex-vertex distance of the pre-cut and post-cut ablated interface was measured manually (in a blind procedure) using ImageJ at $n = 3$ frames after ablation (typically 3-4 s), as previously explained in the Methods section. We thus temporally cropped the original movie in order to limit it to the frames we used for velocity measurements. What happens after these frames does not bring any information about tension prior to ablation. The new movie (Supplementary video 18) is thus clearer because we focused on the essential time-scale and used color bars to indicate the vertex/vertex distance before and immediately after laser ablation.

Overall the influence of tissue wide mechanics in the process of cell movement needs further validation.

Minor points:

- It was a bit annoying that no panels (a,b,c) were stated for Supplementary Figures in the text and did cost some time to find the respective panel referred to, this should be changed.

The panel labelling of Supplementary Figures is now indicated in the main text.

- In Figure 1 it is hard to imagine how single cells could be tracked in 3D when all surrounding cells are labelled. A representative tracking movie featuring tracked cells from the three different areas should be added.

3D tracking of individual *ngn1:gfp+* cells was performed using the labelling of cell nuclei with H2B-RFP, or the labelling of membranes with mbCherry. We added a movie showing an example of an anterior placode cell being manually tracked and followed in X,Y and Z over time (Supplementary Video 2).

- What is the time resolution (and thereby the Δt of MSDs) of imaging described in Figure 1? It would be an advantage to also show standard MSD graphs for clarity, not only the log/log plot.

The images shown in Figure 1 are extracted from Supplementary Movie 1. The Δt between each frame is 10 min. This Δt was the same for all our live imaging analysis of cell movements, in wild type and perturbed conditions. This information is now clearly given in the Methods/live imaging section of the revised manuscript (lines 446,447). The standard MSD graph from which the log/log MSD graph was constructed is now shown in Figure 1g.

- In Figure 1 the authors state that cells were tracked in 3D but all graphs show 2D tracks, this should be explained.

As shown in Figure 1d, the OP tissue only slightly increases in depth (Z) during OP morphogenesis. The relevant axis for visualising and analysing OP coalescence are thus X and Y, which is why we showed mostly 2D tracks, even though cells were tracked in 3D, as illustrated in new Supplementary Video 2. In addition, examples of 3D tracks are presented in Supplementary Figure 5, illustrating the relatively small extent of Z movement for placodal cells, as compared with X,Y displacements.

- Figure 2 b,c,d Filopodia are not obvious in the Figures presented. Maybe a zoom in would help. The same applies for Figure 3 a where axon bundles are hard to see.

The images in Figures 2 and 3a are extracted from movies, so it is difficult to improve the quality. In Figure 2, whole axonal protrusions, and not filopodia, are visible and pointed out. To better see the axon bundles shown in Figure 3a we now also show the green channel alone (*ngn1:gfp*) only in Figure 3a'.

- *In supplementary Figure 3a what kind of mean is shown? Is it an arithmetic mean? How many placodes were analysed? What do the error bars represent?*

As asked by the reviewer in his last point, all the bar graphs have been replaced by box and whiskers plots overlaid with all individual data points. We indicate on the figures and/or in their legends the number of analysed placodes or number of cells in all conditions.

- *The authors should state the number of individual experiments/embryos that were used to generate their specific n=x (for example: n=34 cells from 4 embryos/imaging experiments)*

This information is now given directly on Figures or in Figure legends.

- *The authors should explain, why the live imaging was performed at a rather low temperature of 22°C instead of the widely used 28°C.*

Movies were recorded at the temperature of the imaging facility room (22°C). This is now indicated in the Methods/Live imaging section, lines 444,445.

- *It would be good to add a scheme or bright field image of the head of a zebrafish embryo at relevant developmental stages as Figure 1A. That would help non-experts to orient themselves.*

Schematic views of the heads of embryos at 12s and 24s stages have been added in the new Figure 1 (Figure 1a,b).

- *The references for the Kif5c560-YFP construct are inconsistent: refs. 28, 29 are cited in the results section while ref. 32 is cited in the materials and methods section. The refs. 28, 29 are the correct ones.*

We have corrected this mistake.

- *The combination of green and red in many figures is not suitable for colour-blind readers and also does not give the best possible contrast.*

All figures and movies have been changed in order to improve their reading by colour-blind people.

- *The bar graphs used for a lot of data representation are an inferior way of presenting the data. A better way to present it, in my opinion, would be a boxplot analysis with all data points overlaid.*

As requested by the reviewer, all bar graphs have been replaced by box and whiskers plots overlaid with all individual data points.

Reviewer #3 (Remarks to the Author):

Breau et al. observed in detail dynamic movements of olfactory placode (OP) cells and neighboring cells in zebrafish, and found that OP cells first converge towards the center of the placode along the A-P axis and then move laterally to form a spherical cluster. Perturbing the RhoA/Rock/myosin II pathway in combination with mosaic analysis revealed that convergence movements, but not lateral movements, are active, myosin II-dependent cell migration. Furthermore, by quantitative analysis of nuclei deformation and measuring the initial relaxation speed after laser ablation of cell-cell interfaces, the authors demonstrated that A-P compression forces from actively converging OP cells trigger the lateral cell movements.

In addition to the cellular movements of the developing olfactory placode, the authors observed an interesting mode of axon formation, called retrograde axon extension, where the axon elongation coincides with displacement of cell bodies away from axon tips anchored to the brain surface.

The study is very well executed and relies on a multiplicity of approaches to elucidate dynamics of OP morphogenesis and its underlying mechanisms. The quality of the data and the presentation is very high. This paper presents a major advance towards understanding of how spherical olfactory placodes form from the elongated cellular field.

My major concern with this paper is the interpretation and/or the definition of the retrograde axon extension. The authors state that the retrograde axon extension is "original" in the zebrafish olfactory neurons (line 74-75; line 194-195). It is, however, unclear for me what process/mechanism of the axon extension described here is original. As in the Discussion section, there are literatures showing a similar mode of axon extension in the mammalian brain (reviewed by Hatanaka et al., Proc Jpn Acad Ser B Phys Biol Sci., 2016): e.g., cerebellar granule cells form trailing processes during their radial migration, and the trailing processes give rise to axons. If the authors think that the contribution of "passive" displacement of soma to the retrograde axon extension is the original mechanism, the procedure of axon extension (not completely novel) should be described in distinction from the novelty of underlying mechanism, and the term "original" should be used more carefully.

As mentioned in the Discussion, there are other instances of similar retrograde neurite extension in the nervous system. We now include the Hatanaka et al. reference reviewing the knowledge on trailing processes observed during radial migration of cerebellar granule cells (lines 381-384). We agree with the reviewer and replaced the term "original" mechanism by "non-canonical" or "different".

The authors also state that the retrograde axon extension is a passive process (line 239-240; line 242, subheading; Fig 7). However, I think it is not a simple passive process but a combinatorial process of both active and passive modes. Although the perturbation of microtubule polymerization does not affect lateral movements of cell bodies, axon elongation itself requires microtubule polymerization, an active cellular event, which could be induced concomitantly with the passive, non-autonomous, lateral movement of soma.

Our results strongly support a scenario in which retrograde axon extension is driven by extrinsic mechanical forces that either push or pull the cell bodies away from their axon tips attached to the brain surface. The passive process is the lateral displacement of cell bodies. Axons anchored to the brain must initially elongate through pure stretching due to cell body lateral movements, but after initial stretching novel material (membrane, microtubules) must be added to the axon shaft to accommodate growth. We agree with the reviewer that this

addition of novel material is an active process which likely participates in the axon elongation. This is now discussed in the revised manuscript, lines 363-370.

Minor points:

Fig. 1, legend: Number of cells tracked should be described (c, for anterior, central and posterior cells, respectively). Summary of the movements should be (f), not (e).

The number of tracked cells is now given directly on Figures for all our quantitative live imaging analysis. Summary of the movements in the wild type situation is now shown in Supplementary Figure 7 for comparison with the movements in blebbistatin and rockout-treated embryos.

Fig. 5, legend: "et" should read "and".

This typing mistake has been corrected.

Supplementary Fig. 4, legend: Number of cells tracked should be described (d, e, and f).

The number of tracked cells is now given directly on Figures for all our quantitative live imaging analysis.

Is mbCherry different from mCherry? Is this a membrane-targeted version? If so, please explain when it appears the first time in the text.

MbCherry is a membrane-targeted version of mCherry. We now explain this when it first appears in the text (line 168), and in the Methods section line 410.

REVIEWERS' COMMENTS:

Reviewer #1 (Remarks to the Author):

The authors have significantly improved the quality of the manuscript, particularly by adding the DN Rac data. They have answered all my points and I don't have any additional major concerns. I still have two minor cosmetic points. I believe that the manuscript of Breau and Colleagues after correction of these two minor points will be suitable for publication in Nature Communication.

Minor points:

-Line 250 "Surprisingly, Rac-expressing cells dispersed more than controls..." should be replaced by "Surprisingly, DN-Rac expressing cells dispersed more than controls..."

-In the fig 5, the transparent histograms for DN RhoA and DN Rac are difficult to see. I would fill them with a specific color for each of them to help the visualization.

Reviewer #2 (Remarks to the Author):

The authors did an excellent job in improving this manuscript. The current manuscript is extremely well written and the additional experiments (especially for nuclear shape analysis and axon cutting) and Figure amendments (schemes, plots, colour scheme, etc.) massively help the clarity of the presented results. The authors also did very well on commenting on my previous concerns and explaining how experiments were carried out and why some technical difficulties occurred. While it would have still been nice to have some additional genetic manipulations, I do understand that difficulties can arise depending on the tissue studied. However, the drug experiments are now much better controlled and the results are more convincing.

I only have some remaining minor comments that would only need some text adaptation.

- I still think that it has not absolutely been ruled out that some of the lateral displacement could also bear an active component. Thus, I would rephrase the abstract line 32/33 to 'whereas lateral displacements of cell bodies are of mainly passive nature'.

- It is slightly confusing that in the manuscript the Videos start with Video 3. Maybe this order can be changed.

- I find the data on the axon cutting very convincing and interesting. I noted, that in both conditions the axonless cell bodies show higher MSDs than the ones without axons as if these would be constrained. In case this effect is significant, the authors could maybe comment on it in the discussion.

- As this manuscript comes with many videos, it would help if the authors could add one frame stating what is shown in the video at the beginning.

Congratulations to all authors to this very interesting piece of work.

Reviewer #3 (Remarks to the Author):

All my concerns in the original version of manuscript have been satisfactorily addressed.

Minor comments on the revised manuscript:

1) The authors should state more carefully the results of DN RhoA/DN Rac experiments (lines 223-263). The term DN is sometimes omitted in the text, probably confusing the reader: e.g., line 227, "many RhoA-expressing cells" should read "many DN RhoA-expressing cells"; line 250, "Rac-expressing cells" should read "DN Rac-expressing cells". In addition, the authors inconsistently use the terms "DN RhoA-expressing cells", "DN RhoA cells" and "DN-RhoA" (without "cells"). For the repetitive use, a concise and precise term such as "DN-RhoA+ cells" would be better. The same is true of DN Rac. Please also check the legend of Fig. 5.

2) lines 107, 109, "XY plan" should read "XY plane".

Response to reviewer's comments

We thank the reviewers for the positive assessment of our work, and for catching important typing mistakes.

Reviewer #1

The authors have significantly improved the quality of the manuscript, particularly by adding the DN Rac data. They have answered all my points and I don't have any additional major concerns. I still have two minor cosmetic points. I believe that the manuscript of Breau and Colleagues after correction of these two minor points will be suitable for publication in Nature Communication.

Minor points:

-Line 250 "Surprisingly, Rac-expressing cells dispersed more than controls..." should be replaced by "Surprisingly, DN-Rac expressing cells dispersed more than controls..."

This has been corrected.

-In the fig 5, the transparent histograms for DN RhoA and DN Rac are difficult to see. I would fill them with a specific color for each of them to help the visualization.

We changed the color of the DN-RhoA and DN-Rac histograms and dot distribution plots from white/transparent to orange, which gives a good contrast with the dark grey color used for the control condition. As orange was the only color giving such a good contrast, we used the same color for DN-RhoA and DN-Rac.

Reviewer #2

The authors did an excellent job in improving this manuscript. The current manuscript is extremely well written and the additional experiments (especially for nuclear shape analysis and axon cutting) and Figure amendments (schemes, plots, colour scheme, etc.) massively help the clarity of the presented results. The authors also did very well on commenting on my previous concerns and explaining how experiments were carried out and why some technical difficulties occurred. While it would have still been nice to have some additional genetic manipulations, I do understand that difficulties can arise depending on the tissue studied. However, the drug experiments are now much better controlled and the results are more convincing.

I only have some remaining minor comments that would only need some text adaptation.

- I still think that it has not absolutely been ruled out that some of the lateral displacement could also bear an active component. Thus, I would rephrase the abstract line 32/33 to 'whereas lateral displacements of cell bodies are of mainly passive nature'.

The sentence in the abstract has been rephrased as suggested by the reviewer.

- It is slightly confusing that in the manuscript the Videos start with Video 3. Maybe this order can be changed.

The order of the first 3 movies has been changed accordingly.

- I find the data on the axon cutting very convincing and interesting. I noted, that in both conditions the axonless cell bodies show higher MSDs than the ones without axons as if these would be constrained. In case this effect is significant, the authors could maybe comment on it in the discussion.

As spotted by the reviewer, MSDs tend to be higher in axonless conditions (both laser ablation and colcemid treatment) than in the control situation (with axons), however this slight increase is not statistically different. We therefore chose not to comment on this observation in our manuscript.

- As this manuscript comes with many videos, it would help if the authors could add one frame stating what is shown in the video at the beginning.

This has been done.

Congratulations to all authors to this very interesting piece of work.

Reviewer #3

All my concerns in the original version of manuscript have been satisfactorily addressed.

Minor comments on the revised manuscript:

1) The authors should state more carefully the results of DN RhoA/DN Rac experiments (lines 223-263). The term DN is sometimes omitted in the text, probably confusing the reader: e.g., line 227, "many RhoA-expressing cells" should read "many DN RhoA-expressing cells"; line 250, "Rac-expressing cells" should read "DN Rac-expressing cells". In addition, the authors inconsistently use the terms "DN RhoA-expressing cells", "DN RhoA cells" and "DN-RhoA" (without "cells"). For the repetitive use, a concise and precise term such as "DN-RhoA+ cells" would be better. The same is true of DN Rac. Please also check the legend of Fig. 5.

Changes in the text have been made according to the reviewer's comment.

2) lines 107, 109, "XY plan" should read "XY plane".

This has been corrected.